# Lysosomal trafficking mediated by Arl8b and BORC promotes invasion of cancer cells that survive radiation

Ping-Hsiu Wu [1], Yasuhito Onodera [1,2✉], Amato J. Giaccia[3], Quynh-Thu Le[3], Shinichi Shimizu [1,4], Hiroki Shirato[1] & Jin-Min Nam [1✉]

Enhanced invasiveness, a critical determinant of metastasis and poor prognosis, has been observed in cancer cells that survive cancer therapy, including radiotherapy. Here, we show that invasiveness in radiation-surviving cancer cells is associated with alterations in lysosomal exocytosis caused by the enhanced activation of Arl8b, a small GTPase that regulates lysosomal trafficking. The binding of Arl8b with its effector, SKIP, is increased after radiation through regulation of BORC-subunits. Knockdown of Arl8b or BORC-subunits decreases lysosomal exocytosis and the invasiveness of radiation-surviving cells. Notably, high expression of *ARL8B* and BORC-subunit genes is significantly correlated with poor prognosis in breast cancer patients. Sp1, an ATM-regulated transcription factor, is found to increase BORC-subunit genes expression after radiation. In vivo experiments show that ablation of Arl8b decreases IR-induced invasive tumor growth and distant metastasis. These findings suggest that BORC-Arl8b-mediated lysosomal trafficking is a target for improving radiotherapy by inhibiting invasive tumor growth and metastasis.

[1] Global Center for Biomedical Science and Engineering, Faculty of Medicine, Hokkaido University, 060-8638 Sapporo, Hokkaido, Japan. [2] Department of Molecular Biology, Faculty of Medicine, Hokkaido University, 060-8638 Sapporo, Hokkaido, Japan. [3] Department of Radiation Oncology, Stanford University School of Medicine, Stanford, CA 94305, USA. [4] Department of Radiation Medical Science and Engineering, Faculty of Medicine, Hokkaido University, 060-8638 Sapporo, Hokkaido, Japan. ✉email: yonodera@med.hokudai.ac.jp; jinmini@med.hokudai.ac.jp

Cancer cell invasiveness is associated with metastatic potential[1], which leads to poor prognosis in cancer patients[2]. Previous studies have shown that cell invasiveness is increased in cancer cells that survive cancer treatments such as chemotherapy[3], targeted therapy[4], and radiotherapy[5–7]. Increased invasiveness potentially contributes to resistance and recurrence after these treatments[8,9]. However, the underlying molecular mechanisms by which cancer cells acquire invasiveness after these treatments have not been fully elucidated. Thus, identifying the key molecular mechanisms by which the invasiveness of surviving cancer cells is enhanced after therapy may provide a novel target to improve treatment outcomes.

Cancer invasion is associated with invadopodia formation and matrix degradation by secreted proteases, which lead to microenvironment remodeling[10–13]. Our previous study suggested that breast cancer cells remodel their extracellular matrix (ECM) by secreting fibronectin to aid in survival after ionizing radiation (IR) treatment[14]. Furthermore, using three-dimensional (3D) cultures of a laminin-rich extracellular matrix (lrECM) ductal carcinoma in situ model, we showed that in situ breast cancer cells that survive IR treatment emerge with an invasive phenotype and exhibit upregulated α5β1-integrin expression and fibronectin and matrix metalloprotease (MMP)-9 secretion[5]. IR can alter the cancer microenvironment via lysosome-associated proteins, such as cathepsins and MMPs, which are secreted by cancer cells[8,15].

We and others have shown that vesicle trafficking, including endocytosis, recycling, and the exocytosis of proteins and organelles, plays essential roles in cancer invasion[16–19]. Consistent with this notion, recent studies have highlighted the important roles of lysosomes in tumor biology[20]. In addition to the well-known function of lysosomes as degradative organelles, they are involved in many other cellular processes, including cell adhesion, tumor invasion, and metastasis[20]. These processes are regulated, in part, by lysosomal trafficking toward microtubule minus ends or microtubule plus ends[21]. Lysosomes dock to and fuse with the plasma membrane, after which they discharge their contents[22], including proteolytic enzymes, such as cathepsins or MMPs, into the extracellular space, which leads to ECM remodeling and cancer cell invasion[23]. A recent study provided direct evidence of the close relationship between lysosomal exocytosis and cell invasion by examining invasive protrusion formation in the *Caenorhabditis elegans* anchor cell[24].

The Arf-like small GTPase Arl8b is known as a crucial regulator of lysosomal positioning[25]. As with other members of the Arl family, Arl8b cycles between an inactive (GDP-bound) cytosolic conformation and an active (GTP-bound) membrane-bound conformation. The active form of Arl8b localizes primarily on lysosomes, where it regulates lysosomal trafficking to the cell periphery[25]. In the trafficking of lysosomes, the active form of Arl8b mediates membrane recruitment of the effector protein SifA and kinesin-interacting protein (SKIP, also known as PLEKHM2), which in turn facilitates downstream events to connect lysosomes to kinesin 1[25,26]. Biogenesis of lysosome-related organelles complex 1 (BLOC-1)-related complex (BORC) is required for the activation of Arl8b/SKIP to promote lysosome transport[27]. BORC consists of several subunits, including BLOS1, BLOS2, Myrlysin (LOH12CR1) and others, which mediate the recruitment of Arl8b/SKIP to kinesin, following which the complex promotes lysosomal transport toward the cell periphery[27,28]. Thus, anterograde trafficking of lysosomes from the microtubule-organizing center toward the cell periphery is regulated by the BORC/Arl8b/SKIP complex, which is recruited to kinesin family members[21].

IR exposure induces a series of cellular processes through the activation of transcription factors that regulate the expression of specific genes[29]. Transcription factors are activated by DNA damage sensor proteins, such as ataxia-telangiectasia mutated protein (ATM), ATM and RAD3-related protein (ATR), and DNA-dependent protein kinase (DNA-PK), after IR-induced DNA damage occurs[29]. Sp1 is a transcription factor that was reported to be activated in an ATM-dependent manner[30,31]. While activation of Sp1 is known to regulate the expression of genes related to cancer progression[32,33], the role of Sp1 in lysosomal activation has not yet been reported.

Here, we show that Arl8b-dependent lysosomal exocytosis plays pivotal roles in the enhanced invasiveness of cells that survive IR. By blocking lysosomes with lysosomal inhibitors, IR-induced invasiveness could be suppressed. Lysosomes were distributed to the cell periphery by IR stimulation, which was accompanied with increased lysosomal exocytosis. Arl8b was increased in the lysosomal fraction of IR-surviving (IR-S) cells. Knockdown of Arl8b decreased IR-dependent lysosomal exocytosis and invasion. In addition, we found that the binding of Arl8b to SKIP, which is mediated by BORC, was increased after IR treatment. Moreover, the activation of Sp1 increased the transcription of BORC-subunits after IR. Finally, Arl8b silencing suppressed the increased tumor growth and distant metastasis of IR-S cells in a mouse xenograft model. Our findings suggest a novel mechanism by which the invasiveness of cancer cells that survive radiotherapy is enhanced and may provide a therapeutic strategy to improve cancer treatment.

## Results

**Lysosomes are involved in the invasion of IR-S cancer cells.** Invasiveness can be enhanced in surviving cancer cell population after IR[5]. Recently, lysosomes were implicated in cancer cell invasiveness[20]. To investigate whether lysosomes are involved in the enhanced invasion of IR-S cells, we performed invasion assays with the breast cancer cell lines MDA-MB-231 and Hs578T, which were treated with the lysosomal inhibitors bafilomycin A1 (Baf A1; 4 nM) or chloroquine (CQ; 30 μM) for 12 h with or without IR. The inhibitors suppressed the IR-induced increase in invasiveness in both cell lines (Fig. 1a, b) but did not affect cell viability during the invasion assay (Supplementary Fig. 1a, b). To confirm the effects of the inhibitors on lysosomal morphology, lysosomes were stained with the markers, LysoTracker Red DND-99, and lysosome-associated membrane protein 1 (LAMP1) (Supplementary Fig. 1c). Abnormal lysosomal structures were observed in cells treated with these inhibitors but not in control cells. Compared to the lysosomes in control cells, the lysosomes in cells after Baf A1 or CQ treatment showed an unclear membrane margin with dilated shapes, indicating lysosomal dysfunction as previously shown[34,35]. These data suggest that lysosomes play a role in the enhanced invasion of IR-S breast cancer cell lines.

Lysosomal exocytosis and positioning have been implicated in cancer cell invasion[23,24]. To examine the effect of lysosomal exocytosis in IR-S cells, we evaluated lysosomal exocytosis with dextran-488 (Supplementary Fig. 2a), which predominantly accumulates in lysosomes[36,37]. To evaluate exocytosis, cells were incubated with dextran-488 for 3 h, after which the culture medium was replaced with dextran-free medium (see "Methods" and Supplementary Fig. 2b). We measured the fluorescence of dextran-488 secreted into the culture medium and found that it was significantly higher in IR-S MDA-MB-231 cells than in untreated cells (Fig. 1c). We confirmed that dextran-488 uptake was not affected by IR within 3 h of dextran treatment (Supplementary Fig. 2c). Cell viability was not affected by IR during this assay (Supplementary Fig. 2d). Lysosomes fuse with the plasma membrane by exocytosis[22]. Therefore, we measured exocytosis by detecting LAMP1 on the cell surface as previously

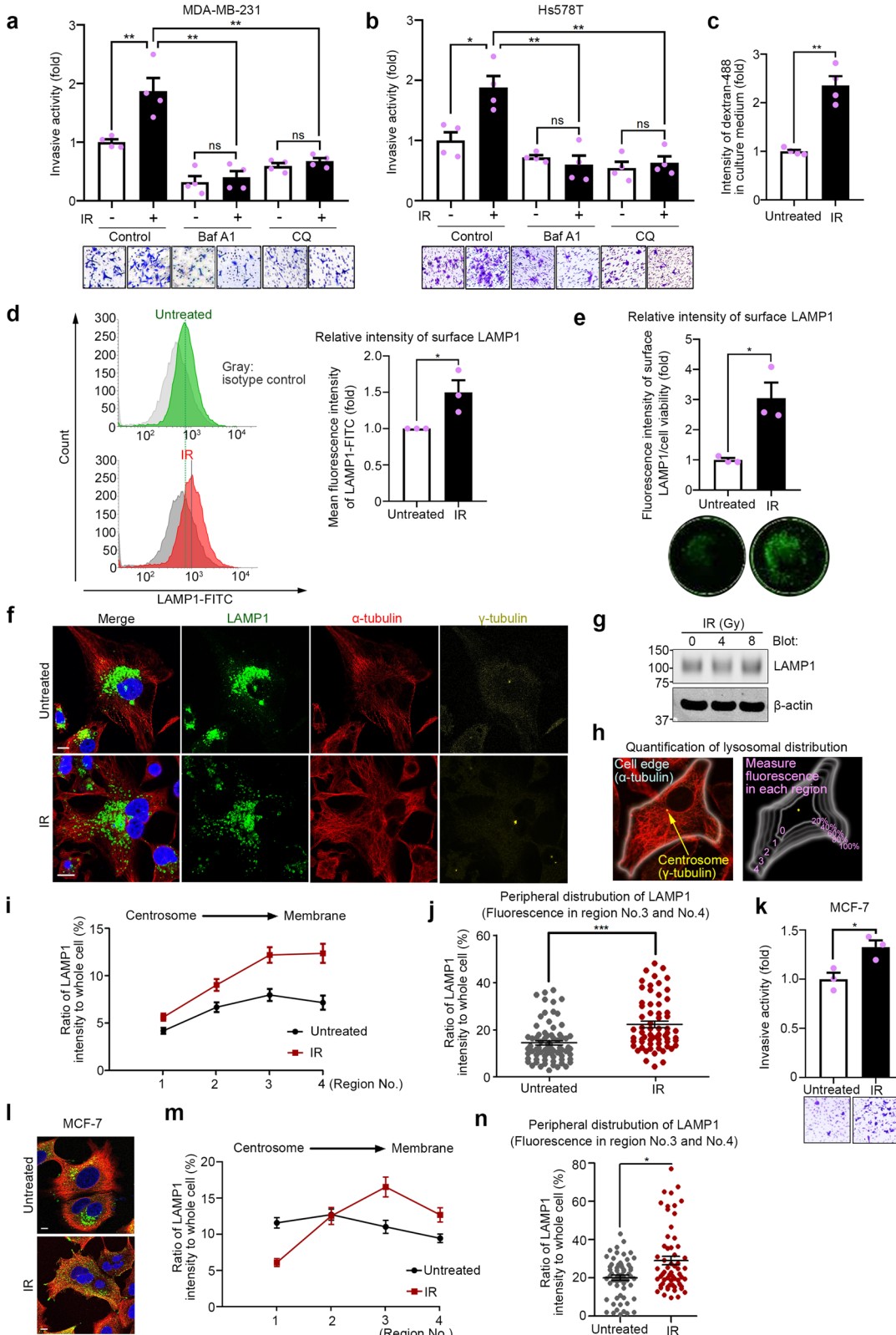

described[38]. MDA-MB-231 cell-surface LAMP1 exposure was significantly increased by IR treatment (Fig. 1d, e), consistent with the dextran-488 release data (Fig. 1c). These data suggest that lysosomal exocytosis is increased in IR-S MDA-MB-231 cells.

Anterograde lysosomal trafficking is essential for lysosomal exocytosis[21]. To further clarify the mechanism of lysosomal exocytosis enhanced in IR-S cells, we analyzed the cytoplasmic distribution of lysosomes by immunofluorescence. LAMP1 staining in IR-treated cells was more diffuse and peripheral (Fig. 1f). In contrast, the lysosomes in untreated cells had a more perinuclear distribution (Fig. 1f). LAMP1 expression was not affected by IR treatment (Fig. 1g). We then quantified the distribution of lysosomes in the same manner previously used to measure mitochondrial distribution[17] (Fig. 1h) and found that the lysosomal

**Fig. 1 Lysosomes are involved in the enhanced invasion of IR-S cancer cells. a, b** Matrigel chemoinvasion assay of MDA-MB-231 cells (**a**) and Hs578T cells (**b**) with or without treatment of lysosomal inhibitors bafilomycin A1 (Baf A1) or chloroquine (CQ) and 4 Gy IR. Representative images of the results were obtained from the Matrigel invasion assays. **c** Lysosomal exocytosis was quantified by the measurement of dextran-488 in culture medium. **d** Cell-surface LAMP1 was measured by flow cytometry. **e** Cell-surface LAMP1 fluorescence intensity was measured using an image scanner. The fluorescence intensity was normalized to the surviving cell viability in each group. **f** Immunofluorescence images of LAMP1 in MDA-MB-231 cells treated with or without IR treatment. Green, LAMP1; red, α-tubulin; yellow, γ-tubulin (centrosome); blue, DAPI (nucleus). Bar, 10 μm. **g** Total LAMP1 in whole-cell lysates of MDA-MB-231 cells after IR treatment was detected by immunoblotting. **h** The lysosomal distribution from the centrosome to the cell membrane was quantified. The cells were divided into regions based on the relative LAMP1 intensity as follows: region 0 (0–20%), region 1 (20–40%), region 2 (40–60%), region 3 (60–80%), and region 4 (80–100%). **i** LAMP1 fluorescence intensities in each region of MDA-MB-231 cells. The intensity of region 0 was excluded. Points and connecting lines, means; bars, SEMs. **j** LAMP1 fluorescence intensity in the cell periphery (regions 3 and 4). More than 20 cells per group were assessed in three independent experiments, and the results are shown as a scatter plot. Bars, SEMs. **k** Matrigel chemoinvasion assay of MCF-7 cells with or without IR treatment. **l** Immunofluorescence images of LAMP1 in MCF-7 cells. Green, LAMP1; red, α-tubulin; blue, DAPI. Bar, 10 μm. **m** LAMP1 fluorescence intensities in each region of MCF-7 cells. **n** LAMP1 fluorescence intensity in the cell periphery (regions 3 and 4) of MCF-7 cells. Bars, SEMs. All column graphs show the means with SEMs of three (**d, e, k**) or four (**a–c**) independent experiments. The results of each group in column graphs were normalized against that of the untreated group. *$P < 0.05$; **$P < 0.01$; ***$P < 0.001$; ns, not significant.

distribution was significantly shifted toward the cell periphery in IR-S cells (Fig. 1i, j). On the other hand, the size and the number of lysosomes were not significantly affected by IR treatment (Supplementary Fig. 2e).

Consistent with the basal subtype breast cancer cell lines (MDA-MB-231 and Hs578T), a luminal A breast cancer cell line, MCF-7, also showed an increase in invasiveness in IR-S cells, which was accompanied by an increase in peripheral lysosomal distribution (Fig. 1k–n and Supplementary Fig. 2f). These results indicate that the enhanced lysosomal exocytosis in IR-S cells is at least partially induced by the increased anterograde trafficking of lysosomes toward the cell periphery.

**Arl8b enhances lysosomal exocytosis and invasiveness.** Anterograde lysosomal trafficking is regulated by the activation of Arl8b[26]. To determine whether Arl8b is involved in the enhanced lysosomal exocytosis in IR-S cells, we first determined the amount of Arl8b in the lysosomal fraction. Arl8b was increased in the lysosomal fraction of IR-S MDA-MB-231 cells compared to untreated cells (Fig. 2a), but the total amount of Arl8b was not affected (Fig. 2b). We then constructed a doxycycline-inducible Arl8b overexpression system (Fig. 2c) to further validate the involvement of Arl8b in lysosomal exocytosis and invasion. Arl8b-mVenus was localized on lysosomes (Fig. 2d), as expected[25]. The increased accumulation of Arl8b on lysosomes by overexpression, as is the case in IR-S cells, induced lysosomal distribution to the cell periphery (Fig. 2e–g). The number of lysosomes in Arl8b-overexpressing cells was not significantly different from that of control cells (Supplementary Fig. 3a). Arl8b overexpression increased chemoinvasion (Fig. 2h), as observed in IR-S cells, but did not affect cell viability (Supplementary Fig. 3b). As mentioned earlier, secreted proteases play an important role in cancer invasion. We determined that essential proteases were localized in Arl8b-positive lysosomes (Fig. 2i). These proteases, including MMP-3, MT1-MMP (MMP-14), cathepsin B and cathepsin D, have been reported to play important roles in cancer invasion, in vivo tumor growth and distant metastasis[39–41].

These results suggest that the increased association of Arl8b with lysosomes can stimulate lysosomal trafficking toward the cell periphery, promoting the invasiveness of cancer cells through protease secretion.

**Arl8b-SKIP association is increased in IR-S cancer cells.** To investigate the roles of Arl8b in the enhanced invasion of IR-S cells, we then focused on Arl8b activity. Arl8b switches between GDP-bound (inactive) and GTP-bound (active) forms; the latter interacts with the effector protein SKIP, which mediates anterograde lysosomal motility[26]. Thus, we determined whether active

Arl8b and its interaction with SKIP were increased in IR-S cells using pulldown assays. The GTP-bound wild-type Arl8b (Arl8b-WT) and the constitutively active GTP-bound mutant Arl8b (Arl8b-Q75L), but not the dominant-negative GDP-bound mutant Arl8b (Arl8b-T34N), colocalized with SKIP (Fig. 3a). Pulldown assays with GST-tagged Arl8b proteins showed that Arl8b-WT and Arl8b-Q75L interacted with SKIP, while Arl8b-T34N only marginally interacted with the effector (Fig. 3b). Interestingly, the binding of Arl8b-WT and SKIP was increased in IR-S cells, suggesting the increased activation of Arl8b in IR-S cells (Fig. 3c). To confirm the activation of endogenous Arl8b in invasive cancer cells, we transfected V5-SKIP and Arl8b-HA into MDA-MB-231 cells. Arl8b-HA was included to balance the expression levels of SKIP and Arl8b, because cell viability was otherwise greatly altered. Arl8b-HA expression was lower than that of the endogenous protein, and these bands could be distinguished by western blotting. As a result, the binding of endogenous Arl8b to V5-SKIP was increased after IR treatment (Fig. 3d). Our data indicate that the activation of Arl8b, which accumulates on lysosomes, is increased in highly invasive cells that survive IR.

**Arl8b is required for the enhanced invasion of IR-S cells.** To further verify the role of Arl8b in the enhanced invasion of IR-S cells, we generated breast cancer cell lines in which Arl8b was knocked down by shRNAs (Fig. 4a and Supplementary Fig. 4a–d). The silencing of Arl8b in MDA-MB-231 cells effectively inhibited the peripheral distribution of lysosomes in IR-S cells (Fig. 4b–d and Supplementary Fig. 4e). Consistent with this difference in lysosomal distribution, the increase in lysosomal exocytosis in IR-S cells was significantly reduced in Arl8b-knockdown cells, as determined by dextran-488 and plasma membrane LAMP1 detection (Fig. 4e–g). Knockdown of Arl8b did not affect the uptake of dextran-488 (Supplementary Fig. 4f). Furthermore, the knockdown of Arl8b prevented the enhanced invasiveness of the IR-S human breast cancer cell lines MDA-MB-231, Hs578T, MCF-7 and the mouse mammary tumor cell line 4T1 (Fig. 4h–k). Notably, Arl8b was required for enhanced lysosomal exocytosis and invasion in IR-S cells but not in untreated cells. Knockdown of Arl8b did not affect cell viability in any of these assays (Supplementary Fig. 4g–k). To determine whether Arl8b affects the long-term survival of cancer cells with or without IR, we performed a clonogenic cell survival assay. Arl8b knockdown did not affect radiosensitivity, as determined by colony formation assay (Supplementary Fig. 4l).

The release of proteases, including the MMP and cathepsin families, is an essential process in ECM degradation and cancer invasion[13]. To assess the mediation of matrix degradation by

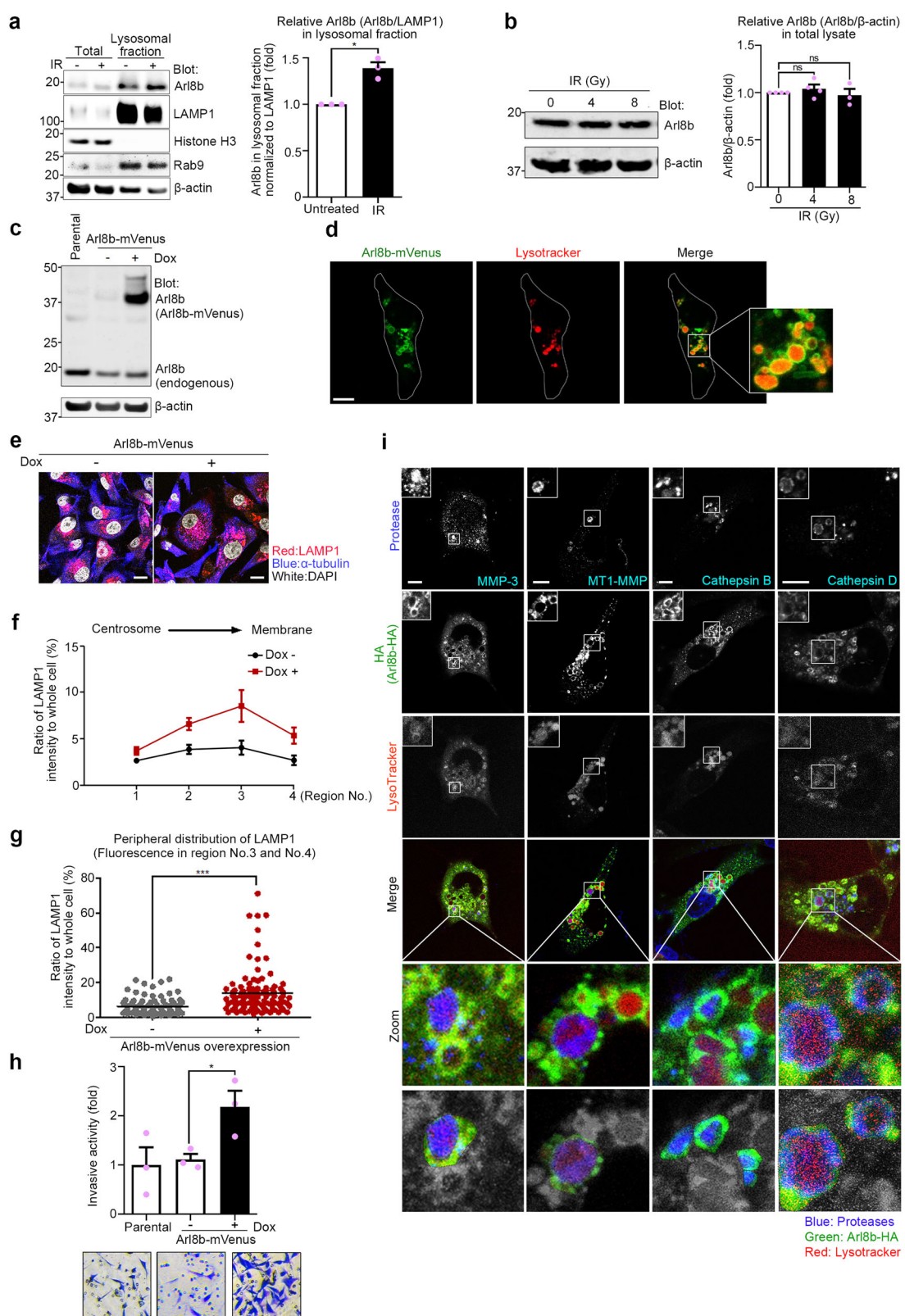

proteases, cells were cultured on lrECM supplemented with fluorescein-conjugated DQ-collagen IV. ECM degradation was increased in IR-S control cells, but this increase in ECM degradation in IR-S cells was suppressed by Arl8b knockdown (Fig. 4l, m). Collectively, these results support the hypothesis that Arl8b plays a key role in lysosomal trafficking to the periphery of

IR-S cells, which in turn results in increased ECM degradation and cancer cell invasion.

**ARL8B and BORC-subunit genes are prognostic indicators**. We next sought to identify the regulatory factors of Arl8b at work in highly invasive cancer cells that survive IR. We focused on the

**Fig. 2 Arl8b on lysosomes enhances lysosomal exocytosis and invasiveness. a** Arl8b in the lysosomal fraction of MDA-MB-231 cells. Markers of the nucleus (Histone H3), lysosomes (Rab9) and the cytoskeleton (β-actin) were stained with antibodies. The amount of Arl8b in the lysosomal fraction was normalized against LAMP1. **b** Total Arl8b in whole-cell lysates of MDA-MB-231 cells after IR treatment was detected by immunoblotting. **c**, **d** Arl8b-mVenus overexpression was induced by doxycycline (**c**) and assessed by confocal fluorescence microscopy (**d**). Dox, doxycycline. Green, Arl8b-mVenus; red, LysoTracker. Bar, 10 μm. **e** Lysosomal distribution in Arl8b-mVenus-overexpressing MDA-MB-231 cells. Pseudocolor micrographs with lysosomes shown in red (LAMP1), microtubules shown in blue (α-tubulin), and nuclei shown in white (DAPI). Bar, 10 μm. **f** LAMP1 fluorescence intensities in each region. The intensity of region 0 was excluded. Points and connecting lines, means; bars, SEMs. **g** LAMP1 fluorescence intensity in the cell periphery (regions 3 and 4). Bars, SEMs. **h** Matrigel chemoinvasion assay of Arl8b-mVenus-overexpressing MDA-MB-231 cells. Representative images of the results were obtained from the Matrigel invasion assays. **i** Immunofluorescence images showing essential proteases in Arl8b-positive lysosomes. Blue, proteases; green, Arl8b-mVenus; red, LysoTracker. Bar, 10 μm. All column graphs show the means with SEMs of three independent experiments. The results of each group in column graphs were normalized against that of the untreated group. *$P < 0.05$; ***$P < 0.001$; ns, not significant.

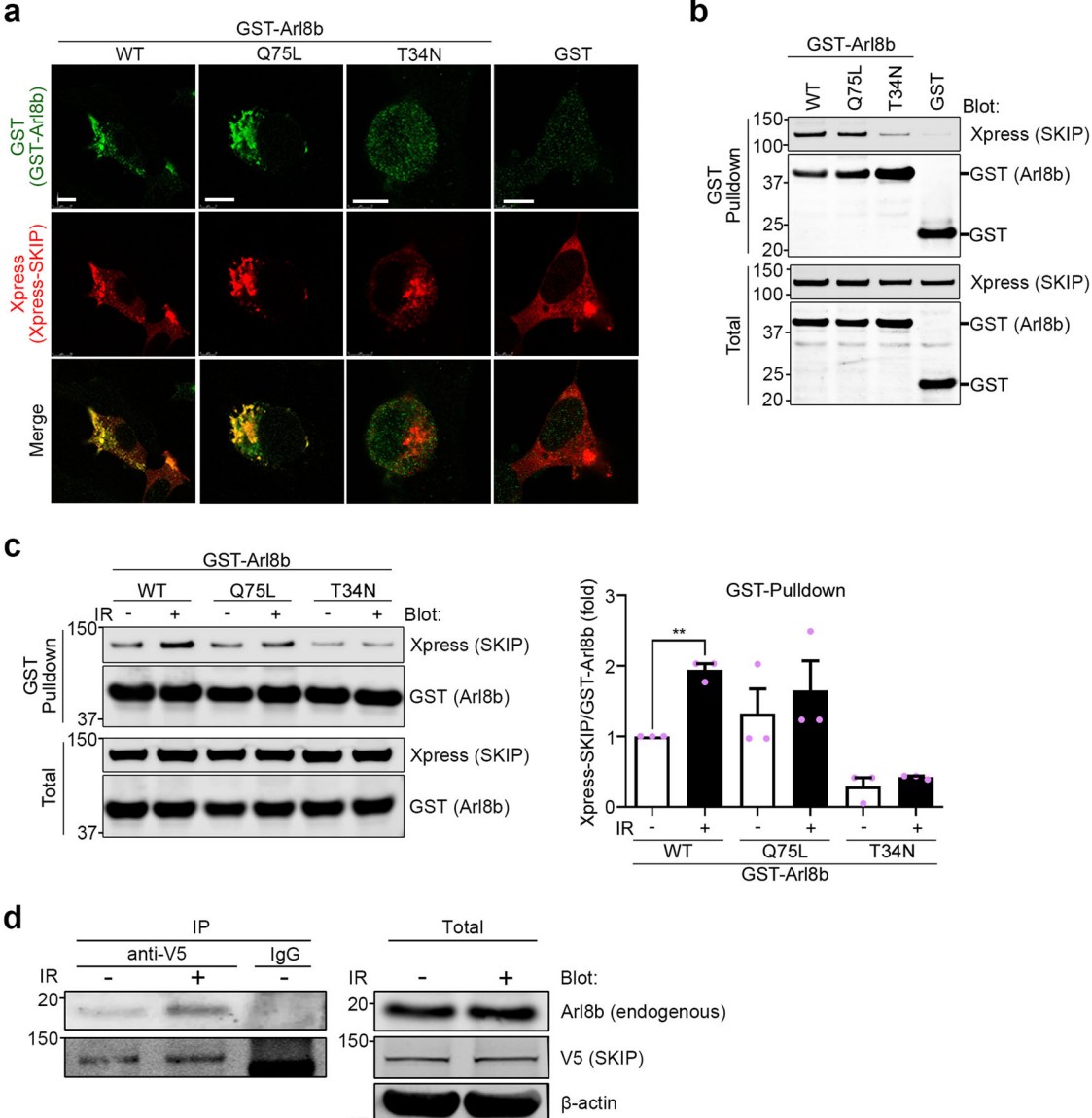

**Fig. 3 The association of Arl8b with its effector is increased in highly invasive cancer cells that survive IR. a** HEK293T cells were transfected with Xpress-SKIP and GST-Arl8b (WT, Q75L or T34N) or GST. Xpress-SKIP and GST-Arl8b were stained with antibodies as indicated. Green, GST; red, Xpress. Bar, 10 μm. **b** GST pulldown assay. **c** GST-Arl8b proteins were pulled down from cell lysates with or without 4 Gy IR treatment. Columns, means ($n = 3$); bars, SEMs. **$P < 0.01. **d** Immunoprecipitation using MDA-MB-231 cell lysates and anti V5-tag antibody beads. MDA-MB-231 cells were transfected with V5-SKIP and Arl8b-HA. IP, immunoprecipitation.

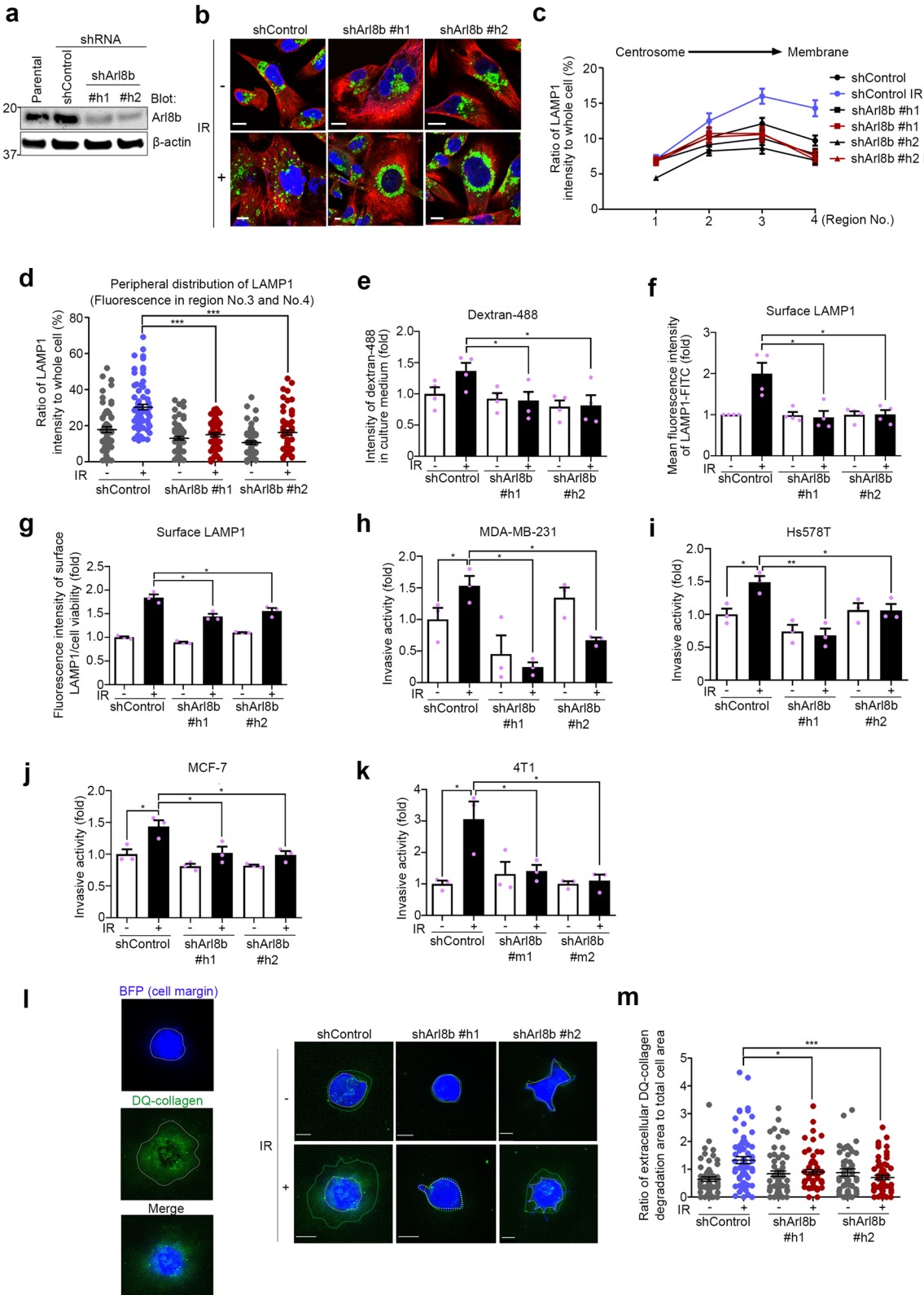

BORC, which was shown to be required for Arl8b recruitment to lysosomes[27]. To determine which subunits of BORC cooperate with Arl8b in cancer progression and invasion, we analyzed a breast cancer dataset available at The Cancer Genome Atlas (TCGA). We found that high *ARL8B* expression is associated with poor patient survival rates (Fig. 5a). Patients expressing high levels of *ARL8B* (*ARL8B*-high patients) were stratified based on

the expression of *BLOC1S1* (BLOS1), *BLOC1S2* (BLOS2), and *BORCS5* (Myrlysin), but not other BORC-subunit genes, further separating the patients with a significantly poor survival rate from other patients, especially in the case of the *ARL8B/BLOC1S2*-high group (Fig. 5a and Supplementary Fig. 5a). The *ARL8B/BLOC1S2*-high group was also found to be associated with an increased rate of lymph node metastasis (Fig. 5b). We

**Fig. 4 Arl8b is required for the enhanced lysosomal exocytosis and invasion of IR-S cancer cells. a** Arl8b was knocked down with shRNAs (shArl8b human #1 and #2) in MDA-MB-231 cells. **b** LAMP1 in control or Arl8b-knockdown MDA-MB-231 cells. Cells were or were not treated with 4 Gy IR. Green, LAMP1; red, α-tubulin; blue, DAPI. Bar, 10 μm. **c** Quantification of the lysosomal distribution in MDA-MB-231 cells. The intensity of region 0 was excluded. Points and connecting lines, means; bars, SEMs. **d** Scatter plot of the lysosomal distribution in the cell periphery (regions 3 and 4). Bars, SEMs. **e** The fluorescence intensities of dextran-488 in the culture medium of control or Arl8b-knockdown MDA-MB-231 cells, with or without IR treatment, were determined with a plate reader. **f** Surface LAMP1 on control and Arl8b-knockdown MDA-MB-231 cells was detected with flow cytometry. **g** Surface LAMP1 on control and Arl8b-knockdown MDA-MB-231 cells was detected with an image scanning system. **h–k** Matrigel chemoinvasion assay of control or Arl8b-knockdown MDA-MB-231 (**h**), Hs578T (**i**), MCF-7 (**j**) and 4T1 (**k**) cells (shArl8b mouse #1 and #2). **l** Matrix degradation activity was determined using a mixture of 3D lrECM and DQ-Collagen IV. Control or Arl8b-knockdown Hs578T cells were cultured in 3D lrECM containing DQ-Collagen IV for 48 h after IR. Blue: blue fluorescent protein (BFP) expressed by cells as a marker of cell margins in live-cell imaging. Green: area of DQ-collagen IV cleaved by proteases. **m** Ratios of extracellular DQ-collagen IV degradation area to the total cell area. Bars, SEMs. All column graphs show the means with SEMs of three (**g–k**) or four (**e, f**) independent experiments. The results of each group in column graphs were normalized against that of the control group. *$P <$ 0.05; **$P <$ 0.01; ***$P <$ 0.001.

then quantified the relative changes in gene expression in 3D-cultured MDA-MB-231 cells after IR treatment by microarray. Among the BORC-subunit genes, *BLOC1S2* and *BORCS5* showed a more than 1.5-fold increase in expression after IR treatment (Fig. 5c). These results suggest that the increased expression and/or activation of Arl8b indeed affects the malignant progression of cancer cells, as is the case for IR-S cells.

Based on these results, we focused on the roles of BLOS2 and Myrlysin in IR-S cells. Knockdown of BLOS2 or Myrlysin suppressed the increase in Arl8b binding with SKIP in IR-S cells (Fig. 5d and Supplementary Fig. 5b). Consistent with a previous study[27], the protein level of Myrlysin was reduced by the knockdown of BLOS2, and vice versa (Supplementary Fig. 5c). Arl8b-HA-transfected MDA-MB-231 cells were used to measure the distribution of Arl8b in cancer cells. BLOS2 or Myrlysin silencing partly suppressed the association of Arl8b-HA and LAMP1-positive lysosomes with or without IR treatment (Fig. 5e). Silencing of BLOS2 or Myrlysin inhibited the increased peripheral distribution of lysosomes and invasion activity in IR-S MDA-MB-231 cells (Fig. 5f–i and Supplementary Fig. 5d), but cell viability was not affected (Supplementary Fig. 5e). These results indicate that BORC-subunits are required for the enhanced invasion mediated by active Arl8b in highly invasive cancer cells that survive IR.

**IR increases BORC gene expression via the ATM-Sp1 axis.** As mentioned in the introduction, IR can regulate gene expression through the activation of transcription factors[29]. Therefore, to determine the mechanism by which IR increases the expression of BORC, we turned our attention to IR-activated transcription factors. Several transcription factors have been reported to be activated after IR treatment, such as Sp1[42], p53[43], and NF-κB[5]. We used ENCODE (www.encodeproject.org) and the JASPAR (jaspar.genereg.net) transcription factor target database to identify transcription factors that may bind to the promoter regions of *BLOC1S2* and *BORCS5*. After comparing the predicted transcription factors with reported IR-activated transcription factors, we focused on Sp1 as a potential target. We found that the knockdown of Sp1 by siRNAs (Fig. 6a) suppressed not only the mRNA expression of *BLOC1S2* and *BORCS5* but also the IR-induced expression of *BLOC1S2* and *BORCS5* (Fig. 6b). We confirmed at the protein level that the phosphorylation of Sp1 at Ser-101, which increased after IR treatment, is partially dependent on ATM (Fig. 6c). ATM is a DNA damage sensor[44] that activates several transcription factors after IR-induced DNA damage[29]. Finally, to confirm that Sp1 increases the DNA transcription of *BLOC1S2* and *BORCS5*, chromatin immunoprecipitation (ChIP)-qPCR was performed. After IR treatment, the binding of Sp1 to the promoters of *BLOC1S2* and *BORCS5* was increased (Fig. 6d). These results indicate that the activation of

Sp1 regulates the increased expression of the BORC-subunits BLOS2 and Myrlysin after IR treatment.

**Arl8b is essential for increased metastasis of IR-S cells.** The association of cancer invasion with tumor growth and metastasis was indicated by recent lysosomal proteases studies, including research on MMP-3[39,45], MT1-MMP[40,46], cathepsin B[47,48], and cathepsin D[49]. Due to the significant effect of Arl8b on highly invasive cancer cells that survive IR, we sought to validate whether Arl8b is required for invasive tumor growth and metastasis in a mouse model. To examine the roles of Arl8b in cells that survive treatment with 4 Gy IR, 4T1-Luc cells stably expressing redshifted *Luciola italica* luciferase were irradiated, following which the IR-S population was selected for 2 days in vitro and then subcutaneously injected into nude mice (Fig. 7a). Consistent with the previous work[50], tumor growth was significantly increased in IR-S cells compared with nonirradiated cells (Fig. 7b, c). However, shRNA-mediated knockdown of Arl8b in 4T1-Luc cells (Supplementary Figs. 4d and 6) effectively suppressed the IR-induced increase in invasive tumor growth in the mice (Fig. 7b, c). To determine the viability of tumor cells in each group, the expression of Ki-67 (a cell proliferation marker) was analyzed by immunohistochemical staining (Fig. 7d). The percentage of proliferating cells (Ki-67-positive cells) was higher in IR-S cells (Fig. 7e). A previous clinical study reported that high levels of Ki-67 positively correlate with a high incidence of lymph node metastasis[51]. We then validated that Arl8b is involved in the distal metastasis of IR-S tumor cells using in vivo imaging and hematoxylin and eosin (H&E) staining of lung tissues. Arl8b knockdown effectively inhibited the IR-induced lung metastasis of IR-S cells in mice (Fig. 7f–h). These results suggested that IR-S cells are characterized by increased tumor growth and metastatic potential, which were suppressed by Arl8b knockdown. Taken together, and consistent with the data obtained in vitro, these in vivo results indicate that Arl8b, a regulator of lysosomal exocytosis, is essential for the increased tumor growth and metastasis of highly invasive cancer cells that survive IR.

**Discussion**

Recent studies have demonstrated possible molecular mechanisms to explain the increased invasiveness of cancer cells that survive IR[5–7]. In this study, we show that increased lysosomal exocytosis in IR-S cells, which may affect the release of proteases, led to increased ECM degradation. Our findings also illuminate the molecular mechanism underlying the increased lysosomal exocytosis in IR-S cells, which is regulated by Arl8b and BORC.

The active form of Arl8b binds its effector, SKIP, to recruit kinesin motors that mediate the transport of lysosomes to the periphery of the cell. The activation of Arl8b appears to be a cellular response to extracellular stimuli and regulates multiple

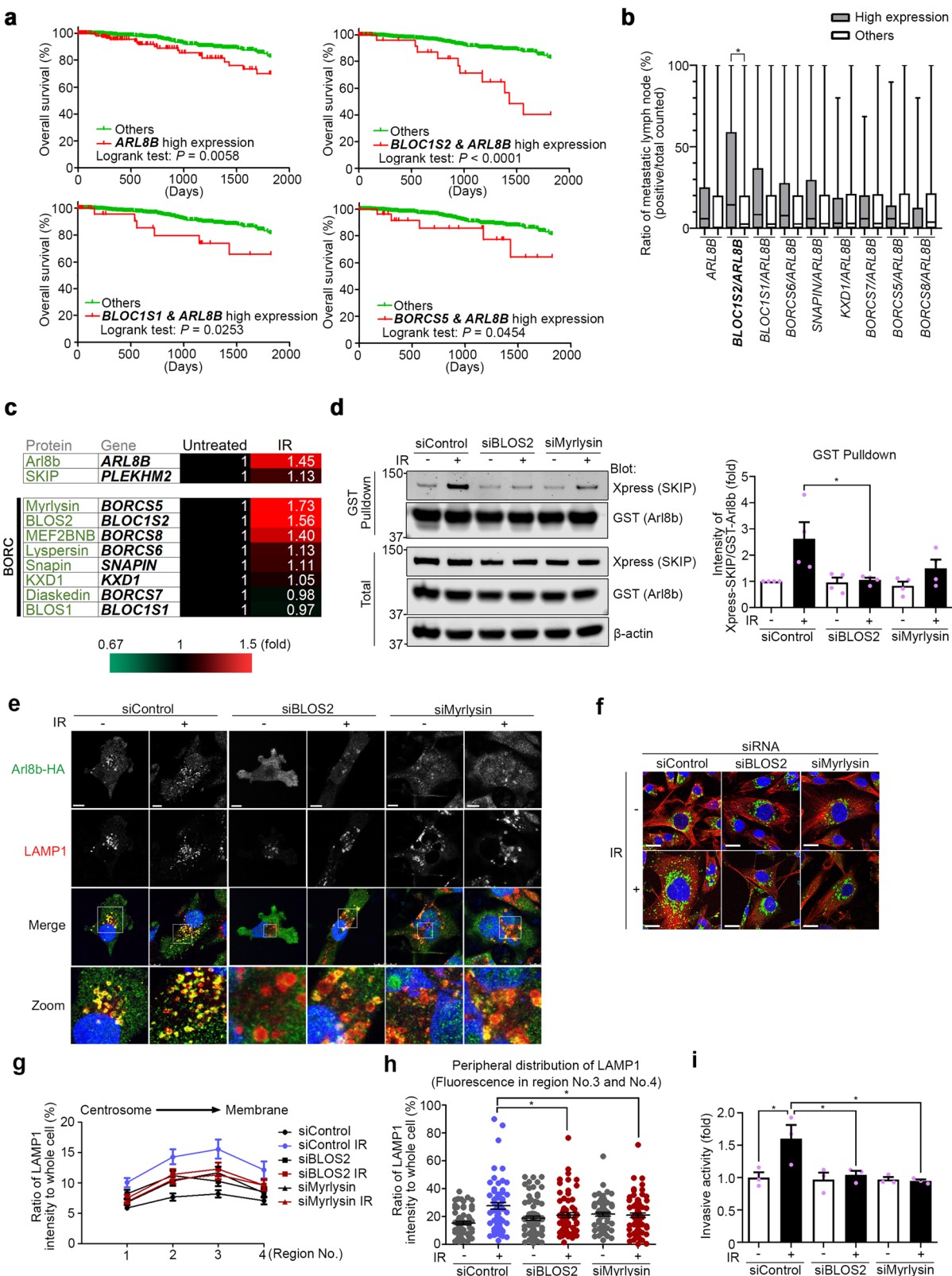

cellular functions. Arl8b facilitates anterograde lysosomal trafficking in response to hepatocyte growth factor, epidermal growth factor, or an acidic extracellular pH, which were shown to be correlated with invasive growth and proteolytic ECM degradation in a 3D model of prostate cancer[52]. Bacterial infection was also shown to activate Arl8b and induce lysosome-mediated plasma membrane repair in HeLa cells and primary macrophages[53]. In addition, the activation of Arl8b by phorbol esters promotes lysosome tubulation in macrophages[54]. In this study, we show that Arl8b activation is increased in highly invasive cancer cells that survive IR and that this process is regulated by BORC. Others' and our data suggest that Arl8b-regulated lysosomal exocytosis is essential for the stimulation-dependent invasiveness, but not the basal invasiveness, of cancer cells.

**Fig. 5 High expression of *ARL8B* and BORC-subunit genes is associated with poor prognosis. a** Kaplan–Meier survival curves were generated based on the expression levels of *ARL8B* and BORC-subunit genes using TCGA data from 1075 human breast cancer patients. **b** Lymph node metastasis (number of positive metastatic lymph nodes/total lymph node count) was found to correlate with the combined expression of *ARL8B* and BORC-subunit genes using the same TCGA data set and stratification as in (**a**). Box-and-whisker plots. **P < 0.01. **c** Heat map of fold-changes in the expression of *ARL8B*, *PLEKHM2* and BORC-subunit genes in MDA-MB-231 cells in 3D lrECM produced by 2 Gy × 4 IR treatment. **d** GST-Arl8b-WT pulldown assays of HEK293T cells transfected with siRNA against BLOS2 or Myrlysin that were then cotransfected with Xpress-SKIP and GST-Arl8b-WT. The Xpress-SKIP signal intensity was normalized by that of GST-Arl8b-WT. Columns, means (*n* = 4); bars, SEMs. *P < 0.05. **e** BLOS2 or Myrlysin was knocked down by siRNA in Arl8b-HA-expressing MDA-MB-231 cells. Green, Arl8b-HA; red, LAMP1; blue, DAPI. Bar, 10 μm. **f** Lysosomal (LAMP1) distribution in MDA-MB-231 cells in which BLOS2 or Myrlysin was knocked down with siRNA. Green, LAMP1; red, α-tubulin; blue, DAPI. Bar, 20 μm. **g** Lysosomal distribution in each region. The intensity of region 0 was excluded. Points and connecting lines, means; bars, SEMs. **h** LAMP1 distribution in the cell periphery (regions 3 and 4). Bars, SEMs. **i** Matrigel chemoinvasion assay of MDA-MB-231 cells treated with siRNA against BLOS2 or Myrlysin. Columns, means (*n* = 3); bars, SEMs. *P < 0.05.

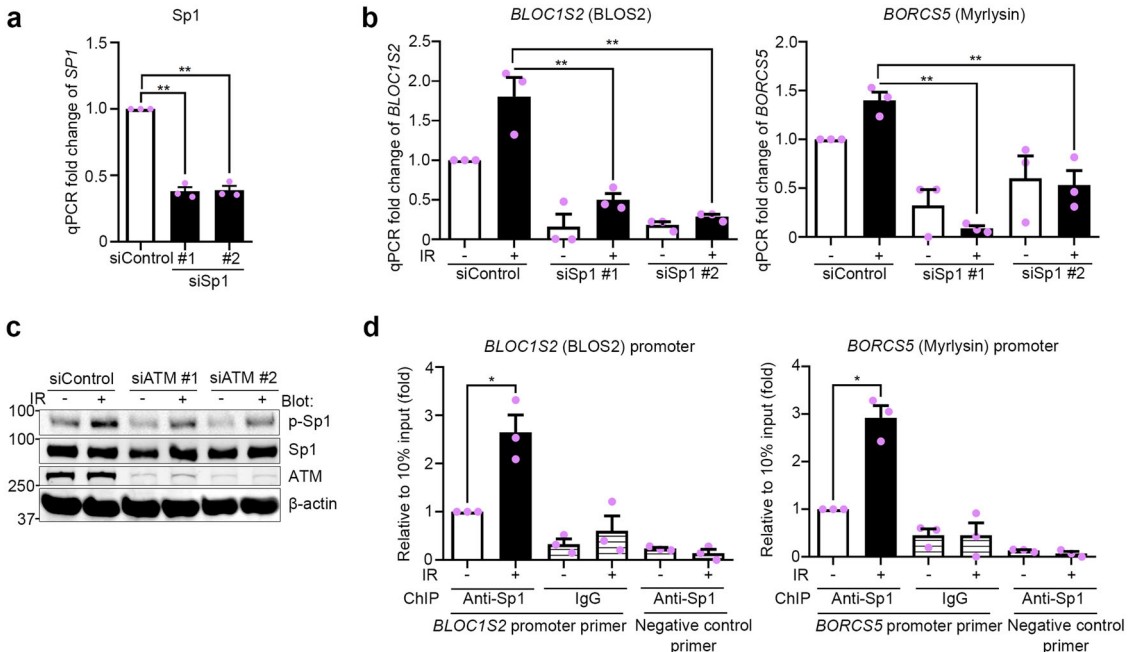

**Fig. 6 IR promotes the expression of BORC-subunit genes by ATM-regulated activation of Sp1. a** Sp1 expression in MDA-MB-231 cells transfected with siRNAs for Sp1 knockdown. 18S rRNA was used as an internal control. **b** Expression of *BLOC1S2* and *BORCS5* mRNA in MDA-MB-231 cells transfected with siRNAs for Sp1 knockdown. 18S rRNA was used as an internal control. **c** Western blot analysis of phosphor-Sp1 (p-sp1) in MDA-MB-231 cells after 4 Gy IR treatment. **d** ChIP-qPCR assays of MDA-MB-231 cells after 4 Gy IR treatment. Immunoprecipitation was performed with anti-Sp1 or non-immune IgG. All column graphs indicate the means with SEMs of three independent experiments. *P < 0.05; **P < 0.01.

Enlarged lysosomes were found in some cells after Arl8b overexpression, as shown in Fig. 2d. The size of the lysosomes may be related to lysosomal activity, but the contribution remains somewhat obscure[55]. Several proteins have been found to affect lysosomal size. For example, the overexpression of TPC[56] or the silencing of MT1-MMP[57] was found to enlarge lysosomes in cells, while the knockdown of Diaskedin[58] or the overexpression of the transcription factor EB[59] reduced lysosomal size. The mechanisms of how these proteins affect lysosomal size are not fully realized. Therefore, the relationship between lysosomal size and these proteins, including Arl8b, is worth further investigation.

Interestingly, we noticed that in some cells after IR treatment, the perinuclear lysosomes lose polarization around the centrosome (Fig. 4b). The distribution of lysosomes can be divided into an immobile pool located around the microtubule-organizing center in a perinuclear "cloud" and a highly dynamic pool in the cell periphery[60]. The cargo contained in the perinuclear cloud, including late endosomes, lysosomes, and trans-Golgi network vesicles, move toward the cell periphery for their following destination[60]. Jongsma et al. showed that the arrest and release of

vesicles from the perinuclear cloud are controlled by an endoplasmic reticulum-located E3 ligase RNF26 and several ubiquitinates[61]. The total release of lysosomes from the perinuclear cloud after IR may be related to the alternation of these proteins, which could be conducted in future studies.

Lysosomes play a critical role in cancer biology[20]. Multiple lysosome-related factors and pathways have been targeted to increase radiosensitivity[62]. Recently, autophagy, a lysosome-mediated degradation process, has gained attention as a target for radiosensitization[63]. A study has shown that cysteine cathepsin proteases are stored in the lysosome lumen, where they contribute to radioresistance[64]. Korolchuk et al. reported that Arl8b activation is dependent on nutrition and suggested that inactivation of Arl8b by starvation induces lysosomal stacking at the centrosomes, increasing the fusion of autophagosomes to lysosomes[65]. In our study, knockdown of Arl8b did not result in a significant difference in colony formation after IR treatment, suggesting that the potential for autophagy, which can be affected by lysosomal position, does not always affect radiosensitivity. Instead, our results highlight the roles of lysosomal trafficking in

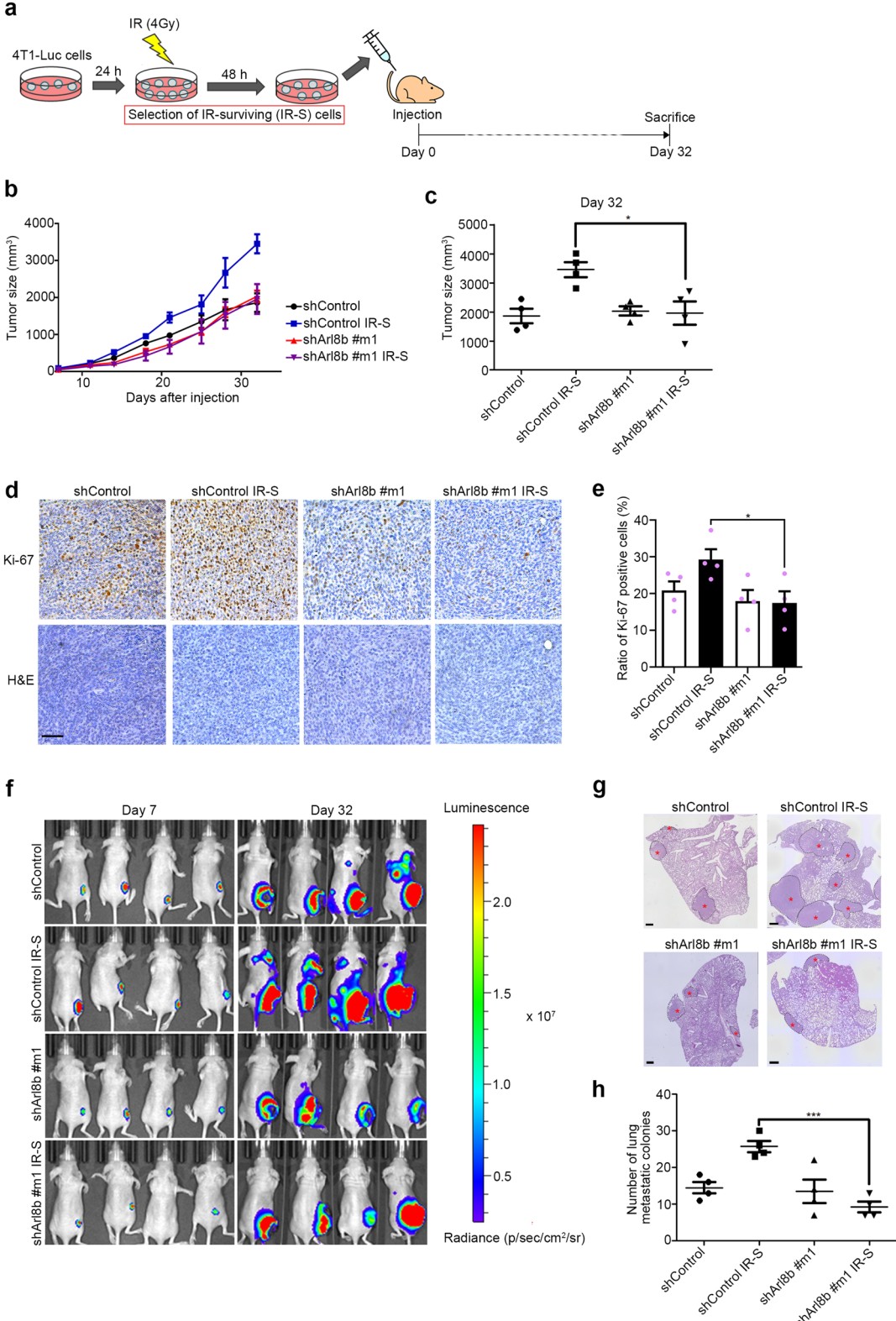

**Fig. 7 Arl8b is essential for the increased tumor growth and metastasis in highly invasive cancer cells that survive IR in vivo. a** Experimental scheme. IR-S cells were selected for 48 h before injection. **b** Tumor growth of 4T1-Luc xenografts in nude mice. Points and connecting lines, means; bars, SEMs. **c** Tumor sizes on day 32. Scatter plot, means (4 mice per group); bars, SEMs. *$P < 0.05$. **d** Sections from 4T1-Luc tumors were subjected to IHC staining with a antibody against Ki-67 (a marker of proliferation). Bar, 100 μm. **e** Ratio of Ki-67 positive cells. Columns, means ($n = 4$); bars, SEMs. *$P < 0.05$. **f** Metastasis of 4T1-Luc cells was detected by luminescence using IVIS. **g** Representative images of H&E-stained lung sections. Red asterisks (*) indicate metastatic nodules. Bar, 500 μm. **h** Analysis of lung metastasis. Scatter plot, means (4 mice per group); bars, SEMs. ***$P < 0.001$.

the invasion of IR-S cancer cells. We also showed that Arl8b silencing suppressed the enhanced invasion of IR-S breast cancer cells in vitro and tumor metastasis in vivo. Analysis of TCGA data also supported our notion that upregulation of the BORC-Arl8b pathway, which is found in highly invasive cancer cells that survive IR, is correlated with increased lymph node metastasis and poor prognosis in breast cancer patients (see also below).

Here, we found that BORC-subunits are required for Arl8b-mediated lysosomal exocytosis and invasion in IR-S cancer cells. Recently, the "Ragulator complex" was reported to negatively regulate the BORC-Arl8b pathway[66,67]. In the presence of nutrients or growth factors, the Ragulator complex is activated and dissociates from BORC, leading to the activation of BORC and the Arl8b-mediated anterograde trafficking of lysosomes[66,67]. Although our data suggest that increased expression of BORC-subunits is involved in the activation of Arl8b, whether dissociation of the Ragulator complex from BORC is also induced by IR should be examined elsewhere. Furthermore, activation of the Ragulator complex is also involved in the regulation of several proteins, including the mechanistic target of rapamycin complex 1 (mTORC1) and Axin, which are involved in the DNA damage response[68–70]. Therefore, the Ragulator complex-related DNA damage response after IR, as well as possible crosstalk within the BORC-Arl8b pathway, will be further investigated in the future.

The distribution of lysosomes to the cell periphery is an important phenotype correlated with cancer invasiveness and aggressiveness[21]. Consistent with this, we found that a more peripheral distribution of lysosomes regulated by Arl8b-BORC-subunits was associated with enhanced invasiveness in breast cancer cells. High expression of both the *ARL8B* and *BLOC1S2* genes, the latter of which is a BORC-subunit, was associated with poor survival rates in breast cancer patients and increased lymph node metastasis, both of which could account for the increased invasiveness. These results indicate that lysosomal positioning regulated by Arl8b-BLOS2 may be correlated with the aggressiveness of breast cancer, irrespective of IR treatment. Recently, Morgan et al. found that highly metastatic cancer cells were preferentially vulnerable to lysosomal inhibition[71]. Lysosomal inhibitors exhibit higher cytotoxicity in highly metastatic bladder cancer cells compared to their less metastatic counterparts[71]. In addition to our data, these studies suggest that lysosomes can be targeted for the therapy of highly metastatic cancer.

In conclusion, our study provides significant insights into how BORC-Arl8b-mediated lysosomal trafficking increases invasiveness in IR-S cells, thereby providing a basis for the development of a novel strategy to improve cancer treatment.

## Methods

**Cell culture and irradiation**. All cell lines were purchased from American Type Culture Collection (ATCC, Manassas, USA) and expanded following their instructions. MDA-MB-231, Hs578T, MCF-7, and HEK293T cells were cultured in DMEM (Sigma-Aldrich, St Louis, USA) containing 10% fetal bovine serum (FBS, HyClone, Logan, USA), while 4T1 mouse mammary tumor cells were cultured in RPMI-1640 (Sigma-Aldrich) containing 10% FBS. To generate 4T1-Luc cells stably expressing redshifted *Luciola italica* luciferase (Red-FLuc)[72], 4T1 cells were transduced with Red-FLuc-carrying lentivirus and selected with 10 µg/ml blasticidin S (Thermo Fisher Scientific, Massachusetts, USA). Cells were irradiated with 130 kV X-rays using a CellRad X-ray generator (Precision X-ray, North Branford, USA).

**Matrigel invasion assay**. The Matrigel chemoinvasion assay was performed using BioCoat Matrigel invasion chambers (Corning, Corning, USA) or 24-well Millicell cell culture inserts with an 8.0 µm PET membrane (Merck Millipore, Burlington, USA) coated with 400 µg/ml Corning Growth Factor Reduced Matrigel basement membrane matrix (Corning) at 37 °C for at least 2 h. Cells were trypsinized after 24 h of IR treatment and loaded into the upper wells of invasion chambers in DMEM without FBS. The lower wells were filled with culture medium containing 10% FBS. MDA-MB-231, Hs578T, MCF-7, and 4T1 cells were loaded in the

chambers and incubated for 5 h, 3 h, 8 h, and 5 h, respectively, at 37 °C in 5% $CO_2$. Cells were fixed in 4% paraformaldehyde (PFA) and stained with 1% crystal violet.

**Immunofluorescence**. To quantify the intracellular lysosomal distribution, MDA-MB-231 cells at 24 h after IR treatment were fixed in a final concentration of 2% PFA by the direct addition of 37% PFA to the culture medium at 37 °C for 10 min. The cells were further fixed and permeabilized with methanol for 5 min at −20 °C and then blocked with 5% bovine serum albumin (BSA) in PBS for 1 h after washing with PBS. Cells were incubated with primary antibodies at a 1:400 dilution in 5% BSA/PBS at 4 °C overnight. After washing, the cells were incubated with Alexa Fluor-conjugated secondary antibodies for 1 h at room temperature. Nuclei were counterstained with 4′,6-diamino-2-phenylindole (DAPI) in PBS for 5 min at room temperature. Immunofluorescence images were acquired using a True Confocal Scanning microscopy system (Leica Microsystems, Wetzlar, Germany).

**Immunoblotting**. The procedure was performed as described previously[6]. Briefly, cells were lysed in radioimmunoprecipitation assay (RIPA) buffer (1% Nonidet P-40 [NP-40] buffer, 150 mM NaCl, 50 mM Tris-HCl [pH 7.4], 5 mM ethylenedia-minetetraacetic acid [EDTA], 1% sodium deoxycholate, 0.1% sodium dodecyl sulfate [SDS], 1 mM $Na_3VO_4$, 1 mM NaF and protease inhibitor cocktail [Merck Millipore]). Cell lysates were aliquoted onto NuPAGE Bis-Tris protein gels (Thermo Fisher Scientific) or SDS-PAGE gels in equal amounts and separated. Proteins were transferred onto polyvinylidene fluoride (PVDF) membranes (Merck Millipore), which were then blocked with Odyssey Blocking Buffer (LI-COR Biosciences, Lincoln, USA) at room temperature for 1 h and subsequently probed with primary antibodies at 4 °C overnight. After washing three times with Tris-buffered saline with Tween-20 (TBST; 25 mM [pH 7.4], 120 mM NaCl, 3 mM KCl and 0.1% Tween-20) for 10 min, the membranes were incubated with secondary antibodies (IRDye 800CW or IRDye 680LT, LI-COR) at room temperature for 1 h. The signals on the membranes were detected using the Odyssey Imager (LI-COR Biosciences).

**Antibodies**. The antibodies used in this study are described in Supplementary Table 1.

**Dextran-488 exocytosis assay**. To quantify exocytosis, $1 \times 10^4$ MDA-MB-231 cells were seeded into 96-well culture plates after 24 h of IR treatment. Cells were treated with Alexa Fluor 488-conjugated dextran at 10,000 MW (Dextran-488, Thermo Fisher Scientific) for 3 h. After washing three times with PBS, 100 µl of culture medium was added to each well. After incubation for 8 h, the culture medium was collected and moved to a new 96-well culture plate. The fluorescence intensity of dextran-488 in the conditioned medium was measured using a microplate reader (CLARIOstar, BMG Labtech, Ortenberg, Germany). Data were collected from three independent experiments in triplicate and normalized against the control group.

**Flow cytometry**. Cells were trypsinized, resuspended in flow cytometry buffer (2% FBS in PBS) and incubated with anti-LAMP1 antibody (Cell Signaling Technology, Danvers, USA; recognizes the luminal/extracellular domain [29-382] of LAMP1) in flow cytometry buffer at 4 °C for 30 min. After washing two times with flow cytometry buffer, cells were stained with Alexa Fluor-conjugated secondary antibodies at 4 °C for 30 min. As an isotype control, cells were stained with secondary antibodies alone. After washing, 10,000 cells per sample were analyzed using a FACSAria III flow cytometer (BD Biosciences, Franklin Lakes, USA). The data were collected and analyzed using BD FACSDiva Software (BD Biosciences). The mean fluorescence intensity of LAMP1 was collected from three independent experiments and normalized against the control group.

**Cell-surface protein staining**. To quantify LAMP1 on the cell surface, cells were cultured in 24-well plates and fixed with 1% PFA without permeabilization. Cells were incubated with primary anti-LAMP1 antibody (Cell Signaling Technology) at 4 °C overnight. After washing with PBS, cells were stained with secondary antibodies (IRDye 800CW, LI-COR Biosciences). Fluorescence intensities were measured with the Odyssey CLx image scanning system (LI-COR Biosciences) and normalized to the cell viability in each group obtained by Cell Counting Kit-8 (CCK-8; Dojindo, Kumamoto, Japan) assay. Data were collected from three independent experiments in triplicate and normalized against the control group.

**Quantification of lysosomal distribution**. To quantify the distribution of lysosomes, multichannel fluorescence images of LAMP1/α-tubulin/γ-tubulin-stained cells were obtained. Each single cell was separated into five regions from the centrosome (defined by γ-tubulin staining) to the cell membrane (defined by α-tubulin staining). The fluorescence intensity of LAMP1 was measured using software, and the ratio of the LAMP1 intensity in each region to the LAMP1 intensity in the whole-cell was calculated as a percentage. For semiautomated measurement of the lysosomal distribution using MetaMorph software (Molecular Devices, San Jose, USA), an original journal macro was developed, as described previously[17].

**Quantification of lysosomal number and size.** The number and size of lysosomes in cells were calculated by analysis of LAMP1-labeled vesicles using MetaMorph software.

**Fractionation of lysosomes.** To enrich the lysosomal fraction of the cells, a Lysosome Enrichment Kit for Tissues and Cultured Cells (Thermo Fisher Scientific) was used according to the manufacturer's instructions. Briefly, cells were trypsinized and resuspended in reagent A. The pellets were vortexed for 5 s and incubated on ice for 2 min. The suspensions were sonicated and treated with an equal volume of reagent B. The mixtures were gently inverted several times and centrifuged at $500 \times g$ at 4 °C for 10 min. Supernatant fractions were collected into new tubes. Gradients (17–30%) were prepared, and a mixture (15%) of supernatant fractions and medium was loaded onto the 17% gradient layer, followed by subcloning into the ultracentrifugation at $145,000 \times g$ at 4 °C for 2 h. The lysosomal fraction was obtained and analyzed by immunoblotting.

**Three-dimensional culture with laminin-rich ECM and DQ-collagen IV.** Matrigel Basement Membrane Matrix (Corning; laminin-rich ECM) was supplemented with 25 µg/ml fluorescein-conjugated DQ-collagen IV to produce 3D matrix culture. Hs578T cells ($2.5 \times 10^4$) transfected with Blue Fluorescent Protein (BFP) were mixed with 130 µl of 3D matrix and seeded into 4 compartments of 35/10 mm CELLview cell culture dishes (Greiner Bio-One, Kremsmünster, Austria). Four hundred microliters of 3D culture medium (DMEM supplemented with 1% FBS, 250 µg/ml insulin, 10 µg/ml transferrin, 2.6 ng/ml sodium selenite, 10–10 M estradiol, $1.4 \times 10^{-6}$ M hydrocortisone and 5 µg/ml prolactin) was added to the top of the 3D matrix. Images of the 3D cultured cells were obtained using a True Confocal scanning microscope system (Leica Microsystems) 2 days after seeding. The areas of cleaved DQ-collagen IV were measured with MetaMorph software. The percentages of the area of cleaved DQ-collagen IV were calculated as follows: (cleaved DQ-collagen IV area − cell area)/cell area. More than 20 cells per group in three independent experiments were analyzed.

**Plasmid construction and transfection.** For Arl8b-mVenus expression, human *ARL8B* cDNA was subcloned into the mVenus N1 vector, followed by subcloning into the PiggyBac transposon-based doxycycline-inducible vector pPB-TRE3G-MCS-CEH-rtTA3-IP[17]. Cotransfection of the resulting Arl8b-mVenus plasmid with the hyperactive PiggyBac transposase (hyPBase) vector[17] into MDA-MB-231 cells was performed using ViaFect™ transfection reagent (Promega, Madison, USA), and cells were then selected by culture with 1 µg/ml puromycin 2 days after transfection. The expression of Arl8b-mVenus was induced by 12.5 ng/ml doxycycline to cause overexpression.

For long-term knockdown, shRNA vectors targeting the sequences described in Supplementary Table 2 were constructed: The oligo-DNAs were cloned into the pEN_TTmiRc2 vector (Addgene, Watertown, USA)[73], followed by subcloning into the PiggyBac transposon-based vector pPB-CEH-MCS-IP with mTagBFP2 cDNA as an expression marker[17]. The cells were cotransfected with the hyPBase plasmid and selected with puromycin as described above.

For GST-Arl8b and Xpress-SKIP expression, *ARL8B* and *PLEKHM2* (SKIP) cDNAs were obtained by PCR from first-strand cDNA synthesized from MDA-MB-231 cells using SuperScript IV reverse transcriptase (Thermo Fisher Scientific). Arl8b-Q75L (a constitutively active mutant deficient in GTP hydrolysis) and Arl8b-T34N (a dominant-negative mutant in which guanine nucleotide exchange factors are blocked) were designed and amplified by PCR. The *ARL8B* cDNAs (wild-type, Q75L, and T34N) were inserted into the pCEHG vector with a GST tag as described previously[17]. The *PLEKHM2* cDNA was cloned into the pcDNA3.1/His C vector (Addgene). Transient transfection into HEK293T cells was performed as described above.

To measure the binding of endogenous Arl8b in MDA-MB-231 cells, V5-SKIP was transfected into MDA-MB-231 cells with Arl8b-HA. To stably express V5-SKIP at a very low level, the pPB PGK IP vector was modified as follows: puromycin-N-acetyltransferase (pac) was moved into the multiple cloning site (MCS), and a new MCS was created after the internal ribosome entry site (IRES) instead, resulting in the pPB-PAC-IRES-MCS vector. A cDNA sequence encoding V5-SKIP, porcine teschovirus-1 2A (P2A) and Arl8b-HA, all of which were connected in frame, was inserted in the MCS of pPB-PAC-IRES-MCS. The resulting vector was stably transfected as described above.

**GST pulldown assay and immunoprecipitation.** For the GST pulldown assay, plasmids containing GST-Arl8b (wild-type or mutants) and Xpress-SKIP were transfected into HEK293T cells as described above. Cells were lysed with NP-40 buffer (1% NP-40, 150 mM NaCl, 20 mM Tris-HCl [pH 7.4], 5 mM EDTA, 1 mM $Na_3VO_4$, 1 mM NaF, and protease inhibitor cocktail [Merck Millipore]) at 24 h after 4 Gy IR treatment. Lysates (200 µg) were incubated with a 10 µl bed volume of glutathione Sepharose 4B beads (GE Healthcare, Chicago, USA) at 4 °C for 2 h with gentle agitation. The beads were extensively washed three times with NP-40 lysis buffer and then incubated at 70 °C for 10 min after the addition of LDS sample buffer (Thermo Fisher Scientific) and 2-mercaptoethanol. Proteins were separated by NuPAGE and analyzed by western blotting.

For immunoprecipitation (IP), cell lysates were incubated with Anti V5-tag Antibody Beads (FUJIFILM, Tokyo, Japan) at 4 °C for 3 h with gentle agitation.

The beads were extensively washed three times with NP-40 lysis buffer and then incubated at 70 °C for 10 min after the addition of LDS sample buffer (Thermo Fisher Scientific) and 2-mercaptoethanol. Proteins were separated by NuPAGE and analyzed by immunoblotting. Immunoblotting of V5 in IP was performed using a rabbit antibody (Cell Signaling Technology) to avoid detection of the mouse antibody beads used for IP. Immunoblotting in total lysate was performed using a mouse V5 antibody (Thermo Fisher Scientific).

**siRNA and transfection.** siRNA duplexes were synthesized by Hokkaido System Science. Cells were transfected with siRNA duplexes using Lipofectamine RNAi-MAX (Thermo Fisher Scientific). The siRNA sequences are listed in Supplementary Table 2.

**Cell viability assay.** Cell viability during the invasion assay or exocytosis assay was determined using the CCK-8 assay. CCK-8 solution was added to cells cultured in 96-well plates for 1 h at 37 °C. To measure cell viability, the optical density at 450 nm was determined using a Multiskan™ GO microplate spectrophotometer (Thermo Fisher Scientific).

**Colony formation assay.** MDA-MB-231 cells were seeded in 6-well plates at different densities: 100 and 200 cells/dish for 0 Gy treatment, 400 and 800 cells/dish for 2 Gy treatment, 4000 and 8000 cells/dish for 5 Gy treatment. On day 14 of culture, the colonies were fixed with 4% PFA for 10 min and stained with 1% crystal violet. Data were collected from four independent experiments and normalized to the colony numbers of the control group.

**TCGA.** RNA-seq and clinical data from breast cancer patients ($n = 1075$) were obtained from The Cancer Genome Atlas. Patients were stratified by *ARL8B* expression; the top 15% of patients with the highest *ARL8B* expression were categorized as the "*ARL8B*-high" group, and the other patients were categorized as the "*ARL8B*-low" group. The *ARL8B*-high group was further stratified by the relative expression levels of each BORC-subunit gene, and the *ARL8B*-high/BORC-high group accounted for 3% of the total patients. Survival curves were estimated based on the Kaplan–Meier method, and survival was compared by logrank test. The same classifications were used to test for correlation with local lymph node metastasis. For each patient, a lymph node metastasis score was calculated as the ratio of positive (by H&E staining) nodes to the total number of examined nodes. Statistical significance was examined by the Brunner–Munzel test, which is independent of distributions and variances.

**Microarray analysis.** MDA-MB-231 cells were cultured on Corning Matrigel Basement Membrane Matrix (Corning) for 3D laminin-rich ECM (lrECM) culture. After IR treatment with 2 Gy × 4 days, total cellular RNA was isolated using a NucleoSpin RNA kit (MACHEREY-NAGEL, Düren, Germany). The high-sensitivity 3D-Gene Human oligo chip 25k v2.10 (Toray Industries, Tokyo, Japan) was used.

**Quantitative real-time PCR.** Total RNA was isolated from MDA-MB-231 cells using TRI Reagent (Sigma-Aldrich). First-strand cDNA was synthesized from the total RNA with SuperScript IV reverse transcriptase (Thermo Fisher Scientific). Quantitative real-time PCR was performed with the LightCycler 96 system (Hoffmann-La Roche, Ltd., Basel, Switzerland) using FastStart Essential DNA Green Master mix (Hoffmann-La Roche, Ltd.). The primers used to amplify target genes are described in Supplementary Table 2.

**ChIP-qPCR.** One to two million MDA-MB-231 cells with or without IR treatment were fixed with 1% formaldehyde for 10 min at room temperature, treated with 1.25 M glycine for 5 min and washed three times with PBS. The cells were collected by scraping, and the dry pellets were flash frozen in liquid nitrogen. The pellets were resuspended in 0.2% NP-40 buffer (0.2% NP-40, 10 mM NaCl, 10 mM Tris-HCl) and incubated for 15 min on ice. The supernatant was discarded, and the dry pellets were resuspended in SDS lysis buffer (1% SDS, 10 mM EDTA [pH 8], 50 mM Tris-HCl [pH 8.1], 1 mM $Na_3VO_4$, 1 mM NaF, and protease inhibitor cocktail; Merck Millipore). After 15 min of incubation on ice, chromatin was sheared to 200–500 bp with sonication (300 W, 10 cycles, 30 s ON/OFF) using a Bioruptor sonicator (UCD-300, Cosmo Bio, Tokyo, Japan). Chromatin samples for immunoprecipitation were incubated with 5 µg of ChIP-grade anti-Sp1 antibody (Cell Signaling Technology) or IgG overnight at 4 °C with agitation. Aliquots containing 1/10 of the chromatin samples were stored at 4 °C for input control. Chromatin and antibody complexes were isolated by incubation with 0.25 µg of Pierce™ Protein A/G magnetic beads (Thermo Fisher Scientific) for 4 h at 4 °C with agitation. The beads were harvested and sequentially washed on a magnetic stand with 1 ml each of the following buffers: low-salt immune complex wash buffer (0.1% SDS, 1% Triton, 2 mM EDTA, 20 mM Tris [pH 8.0], 15 mM NaCl) twice, high-salt immune complex buffer (0.1% SDS, 1% Triton, 2 mM EDTA, 20 mM Tris [pH 8.0], 500 mM NaCl) once, LiCl immune complex buffer (0.25 M LiCl, 1% NP-40, 1% deoxycholic acid, 2 mM EDTA, 20 mM Tris [pH 8.0]) and TE (10 mM Tris, 1 mM EDTA) twice. Chromatin-antibody complexes were eluted using 200 µl of elution buffer (1% SDS, 0.1 M $NaHCO_3$) and 8 µl of 5 M NaCl. Crosslinking was reversed by

incubation of the eluted samples overnight at 65 °C. The supernatant was collected with a magnetic stand. Samples were treated with RNase A for 30 min at 37 °C, followed by proteinase K for 1.5 h at 55 °C. The final samples were purified by phenol-chloroform extraction and ethanol precipitation. Quantitative real-time PCR was performed as described above. The primers used to amplify target genes are described in Supplementary Table 2.

**In vivo study**. To generate 4T1-Luc cells in which Arl8b is stably knocked down, 4T1-Luc cells were transfected with plasmid carrying an Arl8b mouse #1 shRNA sequence. To select IR-S tumor cells, the cells were treated with 0 or 4 Gy IR and incubated for 2 days before tumor injection. 4T1-Luc cells ($1 \times 10^6$ in 100 μl of PBS per mouse) with or without irradiation were subcutaneously injected into the right thighs of 6-week-old female BALB/c-nu/nu mice purchased from Hokudo Co., Ltd. (Sapporo, Japan). The lengths and widths of tumors were measured using a caliper, and the tumor volume was calculated as follows: V = 4/3π × length × width × (length + width)/2 × 1/8. For bioluminescence imaging, the mice were anesthetized with isoflurane and intraperitoneally injected with D-luciferin (Wako, Osaka, Japan) in PBS. Tumor growth and metastasis were detected with bioluminescence using a preclinical in vivo imaging system (IVIS Spectrum CT; Perkin Elmer, Perkin Elmer, USA). For histology, tissues were fixed in a 10% formalin solution (Sigma-Aldrich) at room temperature overnight and incubated in 70% ethanol before being embedded in paraffin. The paraffin blocks were cut into 4-μm sections and mounted onto microscope slides, which were stained with H&E following standard procedures.

For immunohistochemistry, the tumor sections were incubated with antigen unmasking solution (Vector Laboratories, Burlingame, USA) in a 95 °C water bath for 30 min for antigen retrieval. Endogenous peroxidase activity was quenched using 3% $H_2O_2$. Each slide was incubated in 2% blocking buffer (Roche, Basel, Switzerland) for 1 h and then incubated with anti-Ki-67 primary antibody (Cell Signaling Technology) overnight. A Super Sensitive IHC detection system (Biogenex, San Ramon, USA) was used to amplify the signals. Sections were stained with horseradish peroxidase (HRP)-conjugated secondary antibodies. After two washes, the slides were counterstained with hematoxylin (Muto Pure Chemicals, Tokyo, Japan).

All animal experiments were conducted using a protocol approved by the Animal Care and Use Committee of Hokkaido University.

**Statistics and reproducibility**. The in vitro experiments were repeated at least three times. For experiments with a small sample size ($n < 6$), statistical significance was analyzed by two-tailed $t$-test. Graphs are presented as the means ± standard errors of the means (SEMs). For experiments with large sample sizes (analysis of lysosomes), the D′Agostino-Pearson omnibus method was used to test the normality of the data sets, with $P > 0.05$ used to define a normal distribution. For data sets with a normal distribution, statistical significance was examined by $t$-test after confirming equal variance by the $F$-test or $t$-test with Welch's correction if there were significant differences in variance. For data with a non-normal distribution, statistical significance was determined by the Mann–Whitney test. Statistical significance of Fig. 5b was examined by the Brunner–Munzel test, which is independent of distributions and variances. Significant differences are indicated as follows: $*P < 0.05$; $**P < 0.01$; $***P < 0.001$; and ns, not significant. The exact $P$-values are provided in Supplementary Table 3. The experiments were not randomized, but were performed in essentially the same manner. Investigators were not blinded during the experiments and outcome assessment, but had no pre-conceptions.

**Reporting summary**. Further information on research design is available in the Nature Research Reporting Summary linked to this article.

## Data availability
Source data for figures are provided in Supplementary data 1. Uncropped scans of western blot are shown in Supplementary Fig. 7. Microarray data were deposited in the Gene Expression Omnibus (GEO) under accession number GSE155785. RNA-seq and clinical data from breast cancer patients (Project ID: TCGA-BRCA) was obtained at the TCGA web site (URL: http://cancergenome.nih.gov/).

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

## Acknowledgements
We thank Noriko Sasaki and Keiko Kanno for technical assistant; Mari Horikawa, Midori Tokuda, Machiko Kishi, and Nozomi Sato for administrative work. This work was supported in part by the GI-CoRE/GSQ in Hokkaido University, Grant-in-Aid from Ministry of Education, Science, Sports and Culture of Japan (19K08140 to J.N., 18H02759 to Y.O., 19H03591 to H.S.), and a research grant from The Cell Science Research Foundation to Y.O. Analysis of lysosome distribution was supported by Leica Microsystems.

## Author contributions
P.-H.W., Y.O., and J.-M.N. designed the research; P.-H.W., Y.O., A.J.G., Q.-T.L., S.S., H.S., and J.-M.N. performed the research; P.-H.W., Y.O., and J.-M.N. analyzed the data; and P.-H.W., Y.O., and J.-M.N. wrote the paper.

## Competing interests
The authors declare no competing interests.
