## [Peer Review File · Communications Biology]

Reviewers' comments:

Reviewer #1 (Remarks to the Author):

The authors conclude from the manuscript that the invasiveness of radiation-surviving cancer cells is associated with increased lysosomal exocytosis, which is dependent on the activation of Arl8b. The authors show that the radiation-surviving cells exhibit increased lysosomal exocytosis, whereas knockdown of Arl8b or BORG-subunits decreases this and thereby also invasiveness. Besides, the authors correlate high expression of ARL8B and BORG-subunit genes with poor prognosis in breast cancer and perform in vivo mouse experiments to show that Arl8b ablation decreases invasive tumour growth and formation of distant metastases.

We are only just beginning to understand the explicit role of lysosomes in cancer progression, which is why the topic of this study is very interesting and timely. However, my enthusiasm is dampened by the current status of the paper, which I find too preliminary for publication. Specifically, I have concerns on the completeness of the used models, the quality of microscopy experiments and corresponding quantifications, and the depth of the discussion in respect to the current literature. The current set of cell lines should be extended, since both cell lines are triple-negative breast cancer lines. This raises the question if the reported phenotype is specific to this subgroup. Including different breast cancer lines with different subclassifications (ER+, PR+, HER2+) would make the study more complete and increase the impact. For interpretation of the microscopy experiments my main concern is that the authors do not consider differences in lysosomal biogenesis upon radiation of cancer cells. They should analyse if this induces differences in size, number and activity of lysosomes before focusing explicitly on their localization. Radiation could introduce biogenesis of different (yet LAMP-1 positive) endolysosomal organelles via an Arl8b dependent pathway. In relation to that, see Figure 2d, can the authors elaborate on the increased number of LAMP-1 organelles upon overexpression of Arl8b? Finally, some of the data do not fit with existing literature; it is previously reported that cancer cells have more peripheral lysosomes and (mostly) an enhanced lysosomal exocytosis (e.g. PMID: 26921697, PMID: 27105540). According the manuscript however, for example untreated parental MM231 cells have the same level of lysosomal exocytosis as Arl8b knockdown cells (e.g. Figure 3d, e) suggesting lysosomal exocytosis either doesn't occur in parental MM231 cells or is independent of Arl8b. These data are contradictory with the existing literature (e.g. PMID: 26921697, PMID: 27105540). I suggest authors have a critical evaluation of their data on these parts in respect to the existing literature

My detailed comments and experimental concerns are listed below in the order of research questions addressed by each figure of the manuscript:

Lysosomes are involved in the enhanced invasion of IR-S cancer cells.

-12 hours of incubation with lysosomal inhibitors, bafilomycin A1 (Baf A1) and especially with 30uM chloroquine (CQ), is quite long and will probably affect cell viability. Can the authors comment on their choice about the concentration and the duration of incubation with these inhibitors?

-I sincerely doubt the sensitivity of the Cell Counting Kit-8 (CCK-8) assay used to assess viability in these experiments.

-Figure 1a, lower panel crystal violet images are not referred to in the caption.

-Can the authors comment on the change in LAMP-1 labelling after Baf and CQ treatment (Supp. Fig. 1)?

-Figure 1b, is this normalized for viable cell number per condition?

- Figure 1c, this is a quite marginal shift. The authors do not permeabilize the cells for this experiment, but are they certain about the epitope their LAMP-1 antibody (d2d11) recognizes, is it at the N-terminus (luminal/extracellular domain)?

-In 1f, WB shows the same LAMP-1 levels for treated and untreated MM231 cells, and 1g shows IF of LAMP-1. The authors should also quantify the number and size of LAMP-1 labelled organelles besides their distribution. That would provide a more solid indication about the mechanism of lysosome involvement.

-Dextran Uptake experiment (Supp. Fig. 2), no time is indicated for 2a. In 2b and c, The signal intensity is quantified as a measure of uptake. This should be normalized for cell number/ area.

Arl8b on lysosomes enhances lysosomal exocytosis and invasiveness.

- Figure 2a Arl8b/LAMP-1 levels should be corrected for Actin.

- Results presented in 2a and 2b are somewhat contradicting. There seems to be a difference in Arl8b amount in total lysate (2a) if corrected for actin. But in 2b it seems comparable. This should be properly quantified?

- 2c, why are the bands noted as Arl8b-Venus also visible in parental and -dox conditions? This should not be the case. Venus should cause only a ~27kDa shift in the size of Arl8b. A MW marker should be included/shown in this gel?

- The fluorescent lysosomes are very different in size than the lysosomes shown in Figure 1.

- Figure 2d,e: There seems to be more LAMP-1 labelled organelles upon Arl8b overexpression. The authors should add quantification of the total number of organelles per condition, besides the intensity.

- 2f, can you provide the CViolet images as in Figure 1a?

- 2g, colocalization analyses are missing and not visible from the images, in contrast to what authors note in the text and the caption. MMPs, especially MT1-MMP, are previously reported to traffic through a Rab7 positive organelle, not necessarily lysosomes (PMID: 26504170). Also in the figure, there is no direct colocalization between MT1-MMP and Arl8b, Arl8b more seems to be on smaller organelles (vesicles) around MT1-MMP bearing organelles.

Arl8b is required for the enhanced lysosomal exocytosis and invasion of IR-S cancer cells.

- Figure 3b, it is interesting that lysosomal localization not only becomes more peripheral, but also loses polarization around the centromere. Can the authors comment on that?

- It is striking that the parental line shows no difference in collagen degradation compared to the Arl8b knockdown. Can the authors elaborate on that?

The association of Arl8b with its effector is increased in highly invasive cancer cells that survive IR.

- Figure 4a and b can be moved to the supplementary. These show existing information, namely Arl8b binding to its effector SKIP in its active form.

High expression of ARL8B and BORC-subunit genes is associated with poor prognosis.

- (Figure 5) siBLOS2, Arl8b is not recruited to LAMP-1 as expected. But why is it in myristilin kd?

IR promotes the expression of BORC-subunit genes by ATM-regulated activation of Sp1.

Typo in Figure 6d 'negative control

Reviewer #2 (Remarks to the Author):

Evaluation of Manuscript # COMMSBIO-20-0651-T "Lysosomal trafficking mediated by Arl8b and BORC promotes invasion of cancer cells that survive radiation" by Dr Nam and colleagues.

This is a well written and an interesting study about the role of Arl8b and BORC complex in lysosomal trafficking and its role in the invasiveness of radiation surviving cancer cells MDA-MB-231 and Hs578T. After various well controlled in vitro studies authors reach the conclusion that breast cancer cells that survive X-ray irradiation exhibit increased lysosomal exocytosis which is upregulated by Arl8b and which leads into increased invasion of breast cancer cells and may be responsible for metastasis formation and shortened life span of some patients. I think that the study is well done, potentially very important and worth of publishing in Communications biology. I have very little criticism.

Criticism:

-The chapter "Arl8b is required for the enhanced lysosomal..." starting from page 8 is confusing. This is mainly because authors have their story jumping between figures 3 and 4. Authors should either reorganize these two figures or the text so that the story would proceed chronologically from 3a to 4d to make it easier for the readers to follow.

-All western blots are missing molecular weight markers. These should be added.

-Figure 5b requires explanation. Where are these samples from? Lymph node metastasis of what type of cancer? How many samples were used?

Response to Reviewers' Comments

We would like to thank the reviewers for their comments and helpful suggestions to improve our manuscript. Outlined below are our point-by-point responses to the comments (the comments by the reviewers are shown in blue). We have mentioned page and line numbers in some responses. To keep the layout unchanged, please open the manuscript file with Microsoft Word 2013 or higher versions.

Reviewer #1:

The authors conclude from the manuscript that the invasiveness of radiation-surviving cancer cells is associated with increased lysosomal exocytosis, which is dependent on the activation of Arl8b. The authors show that the radiation-surviving cells exhibit increased lysosomal exocytosis, whereas knockdown of Arl8b or BORC-subunits decreases this and thereby also invasiveness. Besides, the authors correlate high expression of ARL8B and BORC-subunit genes with poor prognosis in breast cancer and perform in vivo mouse experiments to show that Arl8b ablation decreases invasive tumour growth and formation of distant metastases.

We are only just beginning to understand the explicit role of lysosomes in cancer progression, which is why the topic of this study is very interesting and timely. However, my enthusiasm is dampened by the current status of the paper, which I find too preliminary for publication. Specifically, I have concerns on the completeness of the used models, the quality of microscopy experiments and corresponding quantifications, and the depth of the discussion in respect to the current literature.

We sincerely thank the reviewer for constructive comments and helpful suggestions. We have added new data and revised the manuscript according to the recommendations, which have significantly improved the paper. For details, please see our point-by-point answers to the reviewer's comments.

***Point 1:** The current set of cell lines should be extended, since both cell lines are triple-negative breast cancer lines. This raises the question if the reported phenotype is specific to this subgroup. Including different breast cancer lines with different subclassifications (ER+, PR+, HER2+) would make the study more complete and increase the impact.*

Response 1: We appreciate the reviewer's suggestion. We agree that it would make the study more complete and increase its impact if we could include a data set with cells of a different subclassification. In addition to the current data set of triple-negative/basal-type breast cancer cell lines, we have also examined cell lines of other

subtypes of breast cancer. Because we focused on the lysosome exocytosis-related molecular mechanism which is essential for "IR-induced invasiveness", we first evaluated these cell lines by performing an invasion assay. After these experiments, we excluded cell lines that showed extremely low invasiveness or did not show radiation-induced invasiveness and found that MCF-7, an ER+/PR+/HER2-/luminal A breast cancer cell line, exhibited IR-induced invasiveness. Therefore, we further analyzed the changes in lysosome distribution and invasiveness in MCF-7 cells after IR treatment and Arl8b knockdown. We found that Arl8b is involved in IR-induced invasiveness in the MCF-7 cell line. These data suggested that Arl8b-mediated lysosomal exocytosis regulates invasiveness after IR not only in triple-negative/basal type cells but also in other subclassifications, such as ER+/PR+/HER2-/luminal A breast cancer cells. We show the results in Figure 1 and Figure 4 of the revised manuscript.

Point 2: For interpretation of the microscopy experiments my main concern is that the authors do not consider differences in lysosomal biogenesis upon radiation of cancer cells. They should analyse if this induces differences in size, number and activity of lysosomes before focusing explicitly on their localization.

Response 2: We understand the reviewer's concern. Therefore, we have quantified the size and the number of lysosomes. The results showed that the size and the number of lysosomes were not significantly affected by IR treatment. We have added these data to Supplementary Figure 2e.

Point 3: Radiation could introduce biogenesis of different (yet LAMP-1 positive) endolysosomal organelles via an Arl8b dependent pathway. In relation to that, see Figure 2d, can the authors elaborate on the increased number of LAMP-1 organelles upon overexpression of Arl8b?

Response 3: We understand that the reviewer thought the total number of "LAMP-1 organelles" was increased by Arl8b overexpression (in Figure 2d) and was therefore concerned about "biogenesis of different (yet LAMP-1-positive) endolysosomal organelles via an Arl8b-dependent pathway". We analyzed the number of LAMP1 organelles and found that it was not significantly increased in Arl8b-overexpressing (doxycycline-activated) cells compared to control (doxycycline-negative) cells. Please see also point 18, which is highly related to this point. These results indicate that Arl8b does not necessarily affect the biogenesis of lysosomes but mainly plays a role in lysosome motility, as also shown in previous studies (PMID: 16537643, PMID: 25898167).

Point 4: Finally, some of the data do not fit with existing literature; it is previously reported that cancer cells have more peripheral lysosomes and (mostly) an enhanced lysosomal exocytosis (e.g. PMID: 26921697, PMID: 27105540). According the manuscript however, for example untreated parental MM231 cells have the same level of lysosomal exocytosis as Arl8b knockdown cells (e.g. Figure 3d, e) suggesting lysosomal exocytosis either doesn't occur in parental MM231 cells or is independent of Arl8b. These data are contradictory with the existing literature (e.g. PMID: 26921697, PMID: 27105540). I suggest authors have a critical evaluation of their data on these parts in respect to the existing literature.

Response 4: We appreciate the reviewer's comment. The lysosomal distribution and exocytosis in cells can be divided into "basal" level (without stimulation) and "stimulation-dependent" level ('stimulation' indicates IR in our study). In our study, we focused on the role of Arl8b in the "stimulation-dependent" level but not the "basal" level. We show that Arl8b knockdown significantly suppressed the IR-induced peripheral lysosomal distribution in IR-surviving cells, but slightly decreased the "basal" level of peripheral lysosomal distribution (Fig. 4c). In our cell lines, the involvement of Arl8b on lysosome exocytosis was clearly evident with regard to the "stimulation-dependent" level rather than the "basal" level.

We can consider two reasons why Arl8b knockdown in MDA-MB-231 cells did not cause a significant suppression of the "basal" level of lysosomal distribution and exocytosis.

1. The "basal" level of peripheral lysosome in MDA-MB-231 breast cancer cells is relatively low. Therefore, Arl8b knockdown in MDA-MB-231 cells did not cause significant changes in lysosomal distribution. However, Arl8b overexpression induces the peripheral distribution of lysosomes, suggesting that Arl8b may be indeed involved in lysosomal exocytosis in MDA-MB-231 cells even without stimulation, although that is not prominent when Arl8b expression is at endogenous levels.
2. The "basal" level of lysosomal distribution may be regulated by not only Arl8b but also other molecules, such as Kif2A/KiF1B (PMID: 21394080), TPM2 (PMID: 23071517), TMEM106b (PMID: 25066864), EIPA (PMID: 19302267), or Neu-1 (PMID: 18606142). The efficacy of Arl8b knockdown may be compromised by other factors.

According to the reviewer's suggestion, we reevaluated our data with regard to the existing literature and confirmed that our data are not contradictory to existing reports. The paper mentioned by the reviewer (PMID: 27105540) clearly showed that knockdown of Arl8b did not affect the "basal" level (red arrows) of lysosomal distribution,

but suppressed the “stimulation-dependent” peripheral lysosomal distribution induced by HGF, EGF or acidic pH (blue arrows), as shown in the figure below. In addition to this study, similar results indicated that the activation of Arl8b after extracellular stimuli (“stimulation-dependent”) has been reported in other studies, which has already been discussed in our original manuscript (Please see page 15, line 316 in the revised manuscript).

(Figure 1b and 1c in PMID: 27105540)

Lysosomes are involved in the enhanced invasion of IR-S cancer cells.

Point 5: 12 hours of incubation with lysosomal inhibitors, bafilomycin A1 (Baf A1) and especially with 30uM chloroquine (CQ), is quite long and will probably affect cell viability. Can the authors comment on their choice about the concentration and the duration of incubation with these inhibitors?

Response 5: In our study, to cause lysosomal destruction without obvious cytotoxicity to cells during the invasion assay, we optimized the treatment conditions through preliminary experiments and finally settled on concentrations of 4 nM Baf A1 and 30 μ M CQ with a 12 h duration. Indeed, the treatment conditions did not affect the viability of MDA-MB-231 cells. Previous studies have also shown that a higher concentration and/or longer treatment periods than our conditions do not affect cell viability (Baf1 A1 [PMID: 22708544]; CQ [PMID: 27916837]).

Point 6: I sincerely doubt the sensitivity of the Cell Counting Kit-8 (CCK-8) assay used to assess viability in these experiments.

Response 6: We appreciate the reviewer for carefully reviewing the Methods section, but we respectfully disagree about this point. Cell counting Kit-8 assay, which was used in our previous study (PMID: 28860745) and others (<https://www.dojindo.com/Cell-Counting-Kit-8>), is a reliable tool to evaluate cell viability. However, we understand the reviewer's concern that the CCK-8 data might not be reliable in our experimental system, although the assay is already established. Therefore, we performed an additional experiment to confirm that the viability data obtained by CCK-8 is essentially same as the data obtained by cell counting (trypan blue dye exclusion) after treatment with lysosome inhibitors and/or IR (Please see the figure below).

The data show no obvious difference in cell viability whether measured by cell counting or the CCK-8 assay in MDA-MB-231 cells, indicating that the CCK-8 data in our study are indeed reliable.

Point 7: Figure 1a, lower panel crystal violet images are not referred to in the caption.

Response 7: We apologize for the lack of detail regarding the lower panel of the crystal violet images. A description of the crystal violet images has been added to the figure legend as follows (page 41, line 921).

“Representative images of the results were obtained from the Matrigel invasion assays.”

Point 8: Can the authors comment on the change in LAMP-1 labelling after Baf and CQ treatment (Supp. Fig. 1)?

Response 8: We thank the reviewer for this suggestion. We have added the following comment to the revised manuscript (page 6, line 116):

“Compared to the lysosomes in control cells, the lysosomes in cells after Baf A1 or CQ treatment showed an unclear membrane margin with dilated shapes (Supplementary Fig. 1c), indicating lysosomal dysfunction as previously shown.”

Point 9: Figure 1b, is this normalized for viable cell number per condition?

Response 9: In the study, we seeded the same number of cells before the exocytosis assay (please also refer to Response 6). Because the cell viability after the assay was not significantly different between each condition (Supplementary Fig. 1), we considered that the cell number did not affect the result, so the results were not normalized to the cell number. The same concept was used in invasion assay in that we did not normalize the results for cell number if the cell viability showed no significant difference during the assay as in our previous study (PMID: 23883667) and others' (PMID: 24962652).

Point 10: Figure 1c, this is a quite marginal shift. The authors do not permeabilize the cells for this experiment, but are they certain about the epitope their LAMP-1 antibody (d2d11) recognizes, is it at the N-terminus (luminal/extracellular domain)?

Response 10: We chose this antibody because the manufacturer (Cell Signaling Technology, CST) guarantees its use in flow cytometry. We inquired of CST what the antigen is, and they confirmed that the #9091 LAMP1 (D2D11) monoclonal antibody was generated using a recombinant protein that consists of the amino acids in the luminal/extracellular domain (29-382) of LAMP1. Therefore, the epitope of the antibody is the luminal/extracellular domain of LAMP1, and the antibody can be used to detect the surface lysosome expression. The above information has been added to Method (page 21, line 465).

Point 11: In 1f, WB shows the same LAMP-1 levels for treated and untreated MM231 cells, and 1g shows IF of LAMP-1. The authors should also quantify the number and size of LAMP-1 labeled organelles besides their distribution. That would provide a more solid indication about the mechanism of lysosome involvement.

Response 11: We thank the reviewer for raising an interesting point. Although our study mainly focused on lysosomal exocytosis on radiation-induced invasion, we agree that the number and size of LAMP1 labeled organelles would be helpful information. Per the reviewer's suggestion, we analyzed the size and number of lysosomes using MetaMorph software and added the data to Supplementary Figure 2e of the revised manuscript. The results showed that the size and the number of lysosomes were not significantly different in cells with or without IR treatment.

Point 12: Dextran Uptake experiment (Supp. Fig. 2), no time is indicated for 2a.

Response 12: Thank you for this comment. The colocalization image was taken after dextran-488 incubation for 3 h. The information has been added to Supplementary Figure 2 and the figure legend of the revised manuscript (Supplementary Figure 2, page 3, line 20 in Supplementary Information).

“Immunofluorescence images showing the colocalization of dextran-488 and LysoTracker. Dextran-488 was incubated for 3 h, and the image was captured after a medium change.”

Point 13: In 2b and c, The signal intensity is quantified as a measure of uptake. This should be normalized for cell number/ area.

Response 13: We thank the reviewer for the suggestion. Regarding Supplementary Figure 2b and c, we have repeated the experiment and normalized the signal intensity to each cell area and revised Supplementary Figure 2c.

Arl8b on lysosomes enhances lysosomal exocytosis and invasiveness.

Point 14: Figure 2a Arl8b/LAMP-1 levels should be corrected for Actin.

Response 14: We have calculated the correlated level of Arl8b and LAMP1 for β -actin in the total lysate as follows (please also see Response 15).

Untreated	IR	
19715	17720	Arl8b
28499	18330	LAMP1
23784	18514	β-actin
0.82892	0.957114	Arl8b/β-actin
1.1982425	0.990062	LAMP1/β-actin

Point 15: Results presented in 2a and 2b are somewhat contradicting. There seems to be a difference in Arl8b amount in total lysate (2a) if corrected for actin. But in 2b it seems comparable. This should be properly quantified?

Response 15: As shown the graph below, the total protein level of Arl8b was not significantly changed after IR treatment. We have included the graph in Figure 2b in the revised manuscript. Furthermore, we also used the Smirnov-Grubbs test to examine whether the normalized value of Arl8b/β-actin in Figure 2a after IR treatment can be an outlier when included in the dataset for the graph. Indeed, the value of Arl8b/β-actin in Figure 2a blot was the most deviated within the dataset, but the T value ($[X - \text{Average}] / \text{SD}$) is less than the critical value for N=4 samples (1.42 for p=0.1, 1.46 for p=0.05, 1.49 for p=0.01), which means that the value of Arl8b/β-actin in Figure 2a blot is NOT an outlier but is in the same population.

Point 16: 2c, why are the bands noted as Arl8b-Venus also visible in parental and -dox conditions? This should not be the case. Venus should cause only a ~27kDa shift in the size of Arl8b. A MW marker should be included / shown in this gel?

Response 16: The band of Arl8b-mVenus was not detected in the parental cell lysate as indicated by the red arrow. In the doxycycline-negative condition, a trace amount of Arl8b-mVenus expression was noted, which could be contributed to the “background expression” of the doxycycline-inducible vector (PMID: 21106052) (PMID: 27216914). Following the reviewer’s suggestion, we have added the molecular weight markers to this image.

Point 17: *The fluorescent lysosomes are very different in size than the lysosomes shown in Figure 1.*

Response 17: We agree with the reviewer that this is an important point. A new paragraph to discuss this point has been added as follows (page 15 line 330).

“Enlarged lysosomes were found in some cells after Arl8b overexpression, as shown in Fig. 2c (ii). The size of the lysosomes may be related to lysosomal activity, but the contribution remains somewhat obscure (PMID: 31808235). Several proteins have been found to affect lysosomal size. For example, the overexpression of TPC (PMID: 23063126) or the silencing of MT1-MMP (PMID: 27478693) was found to enlarge lysosomes in cells, while the knockdown of Diaskedin (PMID: 31314175) or the overexpression of the transcription factor EB (PMID: 26994576) reduced lysosomal size. The mechanisms of how these proteins affect lysosomal size are not fully realized. Therefore, the relationship between lysosomal size and these proteins, including Arl8b, is worth further investigation.”

Point 18: *Figure 2d,e: There seems to be more LAMP-1 labelled organelles upon Arl8b overexpression. The authors should add quantification of the total number of organelles per condition, besides the intensity.*

Response 18: We quantified the total number of LAMP1-labeled organelles, as indicated in Response 3. The data have been added to the revised version of manuscript (Supplementary Fig. 3a). The results showed that the number of lysosomes was not significantly different in Arl8b-overexpressing cells compared to control cells.

Point 19: *2f, can you provide the CViolet images as in Figure 1a?*

Response 19: Thank you for your advice. We have now included representative crystal violet images in Figure 2f of the revised manuscript.

Point 20: 2g, colocalization analyses are missing and not visible from the images, in contrast to what authors note in the text and the caption. MMPs, especially MT1-MMP, are previously reported to traffic through a Rab7 positive organelle, not necessarily lysosomes (PMID: 26504170). Also in the figure, there is no direct colocalization between MT1-MMP and Arl8b, Arl8b more seems to be on smaller organelles (vesicles) around MT1-MMP bearing organelles.

Response 20: We agree on this point, but please note that we did not indicate that the proteases “colocalize” with Arl8b. In Figure 2g, in fact, we show that proteases are contained inside Arl8b-positive lysosomes. This result means that Arl8b does not directly colocalize with these proteases, including MT1-MMP. This finding is also theoretically correct because the proteases are located inside the vesicles, while Arl8b is outside.

Regarding MT1-MMP transport to lysosomes, previous studies have shown the role of lysosome for MT1-MMP delivery (PMID: 31297933, PMID:30111578). Actually, the study provided by the reviewer (PMID: 26504170) does not indicate whether MT1-MMP is transported by lysosomes. Therefore, we think that the study also supports the fact that MT1-MMP could be contained in lysosomes considering the localization of Rab7 on lysosomes (PMID: 10679007).

We realized that our presentation of the data could be misleading. To avoid confusion, we have now included additional images in the bottom of Figure 2g in the revised manuscript.

Arl8b is required for the enhanced lysosomal exocytosis and invasion of IR-S cancer cells.

Point 21: Figure 3b, it is interesting that lysosomal localization not only becomes more peripheral, but also loses polarization around the centromere. Can the authors comment on that?

Response 21: We thank the reviewer for the comments. We have added a short paragraph regarding this finding in the discussion as below (page 16, line 339). Please note that previous Figure 3b is now Figure 4b in the revised version as per the suggestion by Reviewer #2.

“Interestingly, we noticed that in some cells after IR treatment, the perinuclear lysosomes lose polarization around the centrosome (Fig. 4b). The distribution of lysosomes can be divided into an immobile pool located around the microtubule-organizing center in a perinuclear “cloud” and a highly dynamic pool in the cell periphery (PMID: 29900632). The cargo contained in the perinuclear cloud, including late endosomes, lysosomes, and trans-Golgi network vesicles, move toward the cell periphery for their following destination (PMID: 29900632). Jongsma et al. showed that the arrest and release of vesicles from the perinuclear cloud are controlled by an endoplasmic reticulum-located E3 ligase RNF26 and several ubiquitinates (PMID: 27368102). The total release of lysosomes from the perinuclear cloud after IR may be related to the alternation of these proteins, which could be conducted in future studies.”

Point 22: It is striking that the parental line shows no difference in collagen degradation compared to the Arl8b knockdown. Can the authors elaborate on that?

Response 22: We show that Arl8b-mediated invasion and degradation is regulated by lysosomal exocytosis induced by ‘radiation stimulation’. Therefore, in parental MDA-MB-231 cells, Arl8b knockdown does not affect collagen degradation. The fact is consistent with the results of *in vivo* tumor growth.

The association of Arl8b with its effector is increased in highly invasive cancer cells that survive IR.

Point 23: Figure 4a and b can be moved to the supplementary. These show existing information, namely Arl8b binding to its effector SKIP in its active form.

Response 23: In our original version of manuscript, we included the data because we could not find previous reports that GST-tagged Arl8b colocalizes with SKIP (Fig. 3a in revised manuscript), and GST-Arl8b and SKIP pull down its active form in a mammalian cell line (Fig. 3b in revised manuscript) although these finding have been reported using GFP-tag and purified protein systems, respectively. Therefore, we would like to keep these data because the results contain new information and strengthen our manuscript.

High expression of ARL8B and BORC-subunit genes is associated with poor prognosis.

Point 24: (Figure 5) siBLOS2, Arl8b is not recruited to LAMP-1 as expected. But why is it in myrisilin kd?

Response 24: In Figure 5e, Arl8b is obviously disassociated from LAMP1 in siMyrlysin cells compared to the siControl group. As the reviewer noted, the decrease in Arl8b recruitment to LAMP1 in siBLOS2 cells appears clearer than that in siMyrlysin cells. We believe the difference is due to differences in knockdown efficiency in siBLOS2 (approximately 70% decrease) and siMyrlysin (approximately 56% decrease) compared to siControl.

IR promotes the expression of BORC-subunit genes by ATM-regulated activation of Sp1.

Point 25: Typo in Figure 6d 'negative control

Response 25: We apologize for the typographical error. We have corrected the error in the revised manuscript.

Reviewer #2:

This is a well written and an interesting study about the role of Arl8b and BORC complex in lysosomal trafficking and its role in the invasiveness of radiation surviving cancer cells MDA-MB-231 and Hs578T. After various well controlled in vitro studies authors reach the conclusion that breast cancer cells that survive X-ray irradiation exhibit increased lysosomal exocytosis which is upregulated by Arl8b and which leads into increased invasion of breast cancer cells and may be responsible for metastasis formation and shortened life span of some patients. I think that the study is well done, potentially very important and worth of publishing in Communications biology. I have very little criticism.

We thank the reviewer for acknowledgement of the interest and importance of our work. We have revised and improved our manuscript and address the points suggested by the reviewer below.

***Point 1:** -The chapter "Arl8b is required for the enhanced lysosomal..." starting from page 8 is confusing. This is mainly because authors have their story jumping between figures 3 and 4. Authors should either reorganize these two figures or the text so that the story would proceed chronologically from 3a to 4d to make it easier for the readers to follow.*

Response 1: We sincerely thank the reviewer for the suggestion. We have rearranged the order of the chapter "The association of Arl8b with its effector is increased in highly invasive cancer cells that survive IR" (originally Fig. 4) and the chapter "Arl8b is required for the enhanced lysosomal exocytosis and invasion of IR-S cells" (originally Fig. 3). The revised text is shown below. We believe that the reorganization and rearrangement make the article easier for the readers to follow.

"The association of Arl8b with its effector is increased in highly invasive cancer cells that survive IR.

To investigate the roles of Arl8b in the enhanced invasion of IR-S cells, we then focused on Arl8b activity. Arl8b switches between GDP-bound (inactive) and GTP-bound (active) forms; the latter interacts with the effector protein SKIP, which mediates anterograde lysosomal motility...

Arl8b is required for the enhanced lysosomal exocytosis and invasion of IR-S cells.

To further verify the role of Arl8b in the enhanced invasion of IR-S cells, we generated breast cancer cell lines in which Arl8b was knocked down by shRNAs (Fig. 3a and Supplementary Fig. 4a-c). ..."

Point 2: -All western blots are missing molecular weight markers. These should be added.

Response 2: We apologize for the lack of molecular weight markers in the western blots. The molecular weight markers have been added to all western blot results.

Point 3: -Figure 5b requires explanation. Where are these samples from? Lymph node metastasis of what type of cancer? How many samples were used?

Response 3: The samples used in Figure 5b were same as those in Figure 5a described in the “TCGA” section of the Methods. Therefore, the lymph node metastatic results were obtained from breast cancer patients (n=1075) in the TCGA database. We have added an explanation to the figure legends as follows (page 50, line 1014):
“using the same TCGA data set and stratification as in (a).”

REVIEWERS' COMMENTS:

Reviewer #1 (Remarks to the Author):

Authors have added new data and revised their work according to the comments and recommendations I have raised previously. These additions have significantly improved the manuscript.